# LEARNING THE POSITIONS IN COUNTSKETCH

**Yi Li** [*]
Nanyang Technological University
yili@ntu.edu.sg

**Honghao Lin, Simin Liu**
Carnegie Mellon University
{honghaol, siminliu}@andrew.cmu.edu

**Ali Vakilian**
Toyota Technological Institute at Chicago
vakilian@ttic.edu

**David P. Woodruff**
Carnegie Mellon University
dwoodruf@andrew.cmu.edu

## ABSTRACT

We consider sketching algorithms which first compress data by multiplication with a random sketch matrix, and then apply the sketch to quickly solve an optimization problem, e.g., low-rank approximation and regression. In the learning-based sketching paradigm proposed by Indyk, Vakilian, and Yuan (2019), the sketch matrix is found by choosing a random sparse matrix, e.g., CountSketch, and then the values of its non-zero entries are updated by running gradient descent on a training data set. Despite the growing body of work on this paradigm, a noticeable omission is that the locations of the non-zero entries of previous algorithms were fixed, and only their values were learned. In this work, we propose the first learning-based algorithms that also optimize the locations of the non-zero entries. Our first proposed algorithm is based on a greedy algorithm. However, one drawback of the greedy algorithm is its slower training time. We fix this issue and propose approaches for learning a sketching matrix for both low-rank approximation and Hessian approximation for second order optimization. The latter is helpful for a range of constrained optimization problems, such as LASSO and matrix estimation with a nuclear norm constraint. Both approaches achieve good accuracy with a fast running time. Moreover, our experiments suggest that our algorithm can still reduce the error significantly even if we only have a very limited number of training matrices.

## 1 INTRODUCTION

The work of (Indyk et al., 2019) investigated learning-based sketching algorithms for low-rank approximation. A sketching algorithm is a method of constructing approximate solutions for optimization problems via summarizing the data. In particular, *linear* sketching algorithms compress data by multiplication with a sparse "sketch matrix" and then use just the compressed data to find an approximate solution. Generally, this technique results in much faster or more space-efficient algorithms for a fixed approximation error. The pioneering work of Indyk et al. (2019) shows it is possible to learn sketch matrices for low-rank approximation (LRA) with better average performance than classical sketches.

In this model, we assume inputs come from an *unknown* distribution and learn a sketch matrix with strong expected performance over the distribution. This distributional assumption is often realistic – there are many situations where a sketching algorithm is applied to a large batch of related data. For example, genomics researchers might sketch DNA from different individuals, which is known to exhibit strong commonalities. The high-performance computing industry also uses sketching, e.g., researchers at NVIDIA have created standard implementations of sketching algorithms for CUDA, a widely used GPU library. They investigated the (classical) sketched singular value decomposition (SVD), but found that the solutions were not accurate enough across a spectrum of inputs (Chien & Bernabeu, 2019). This is precisely the issue addressed by the learned sketch paradigm where we optimize for "good" average performance across a range of inputs.

---

[*]All authors contributed equally.

While promising results have been shown using previous learned sketching techniques, notable gaps remain. In particular, all previous methods work by initializing the sketching matrix with a random sparse matrix, e.g., each column of the sketching matrix has a single non-zero value chosen at a uniformly random position. Then, the *values* of the non-zero entries are updated by running gradient descent on a training data set, or via other methods. However, the *locations* of the non-zero entries are held fixed throughout the entire training process.

Clearly this is sub-optimal. Indeed, suppose the input matrix $A$ is an $n \times d$ matrix with first $d$ rows equal to the $d \times d$ identity matrix, and remaining rows equal to $0$. A random sketching matrix $S$ with a single non-zero per column is known to require $m = \Omega(d^2)$ rows in order for $S \cdot A$ to preserve the rank of $A$ (Nelson & Nguyên, 2014); this follows by a birthday paradox argument. On the other hand, it is clear that if $S$ is a $d \times n$ matrix with first $d$ rows equal to the identity matrix, then $\|S \cdot Ax\|_2 = \|Ax\|_2$ for all vectors $x$, and so $S$ preserves not only the rank of $A$ but all important spectral properties. A random matrix would be very unlikely to choose the non-zero entries in the first $d$ columns of $S$ so perfectly, whereas an algorithm trained to optimize the locations of the non-zero entries would notice and correct for this. This is precisely the gap in our understanding that we seek to fill.

**Learned CountSketch Paradigm of Indyk et al. (2019).**    Throughout the paper, we assume our data $A \in \mathbb{R}^{n \times d}$ is sampled from an unknown distribution $\mathcal{D}$. Specifically, we have a training set $\mathsf{Tr} = \{A_1, \ldots, A_N\} \in \mathcal{D}$. The generic form of our optimization problems is $\min_X f(A, X)$, where $A \in \mathbb{R}^{n \times d}$ is the input matrix. For a given optimization problem and a set $\mathcal{S}$ of sketching matrices, define $\mathsf{ALG}(\mathcal{S}, A)$ to be the output of the classical sketching algorithm resulting from using $\mathcal{S}$; this uses the sketching matrices in $\mathcal{S}$ to map the given input $A$ and construct an approximate solution $\hat{X}$. We remark that the number of sketches used by an algorithm can vary and in its simplest case, $\mathcal{S}$ is a single sketch, but in more complicated sketching approaches we may need to apply sketching more than once—hence $\mathcal{S}$ may also denote a set of more than one sketching matrix.

The learned sketch framework has two parts: (1) offline sketch learning and (2) "online" sketching (i.e., applying the learned sketch and some sketching algorithm to possibly unseen data). In offline sketch learning, the goal is to construct a CountSketch matrix (abbreviated as CS matrix) with the minimum expected error for the problem of interest. Formally, that is,

$$\arg\min_{\mathsf{CS}\ S} \mathbf{E}_{A \in \mathsf{Tr}} f(A, \mathsf{ALG}(S, A)) - f(A, X^*) = \arg\min_{\mathsf{CS}\ S} \mathbf{E}_{A \in \mathsf{Tr}} f(A, \mathsf{ALG}(S, A)),$$

where $X^*$ denotes the optimal solution. Moreover, the minimum is taken over all possible constructions of CS. We remark that when ALG needs more than one CS to be learned (e.g., in the sketching algorithm we consider for LRA), we optimize each CS independently using a surrogate loss function.

In the second part of the learned sketch paradigm, we take the sketch from part one and use it within a sketching algorithm. This learned sketch and sketching algorithm can be applied, again and again, to different inputs. Finally, we augment the sketching algorithm to provide worst-case guarantees when used with learned sketches. The goal is to have good performance on $A \in \mathcal{D}$ while the worst-case performance on $A \notin \mathcal{D}$ remains comparable to the guarantees of classical sketches. We remark that the learned matrix $S$ is trained offline only once using the training data. Hence, no additional computational cost is incurred when solving the optimization problem on the test data.

**Our Results.**    In this work, in addition to learning the values of the non-zero entries, we learn the locations of the non-zero entries. Namely, we propose three algorithms that learn the locations of the non-zero entries in CountSketch. Our first algorithm (Section 4) is based on a greedy search. The empirical result shows that this approach can achieve a good performance. Further, we show that the greedy algorithm is provably beneficial for LRA when inputs follow a certain input distribution (Section F). However, one drawback of the greedy algorithm is its much slower training time. We then fix this issue and propose two specific approaches for optimizing the positions for the sketches for low-rank approximation and second-order optimization, which run much faster than all previous algorithms while achieving better performance.

For low-rank approximation, our approach is based on first sampling a small set of rows based on their ridge leverage scores, assigning each of these sampled rows to a unique hash bucket, and then placing each non-sampled remaining row in the hash bucket containing the sampled row for which it is most similar to, i.e., for which it has the largest dot product with. We also show that the worst-case guarantee of this approach is strictly better than that of the classical Count-Sketch (see Section 5).

For sketch-based second-order optimization where we focus on the case that $n \gg d$, we observe that the actual property of the sketch matrix we need is the *subspace embedding property*. We next optimize this property of the sketch matrix. We provably show that the sketch matrix $S$ needs fewer rows, with optimized positions of the non-zero entries, when the input matrix $A$ has a small number of rows with a heavy leverage score. More precisely, while CountSketch takes $O(d^2/(\delta\epsilon^2))$ rows with failure probability $\delta$, in our construction, $S$ requires only $O((d \operatorname{polylog}(1/\epsilon) + \log(1/\delta))/\epsilon^2)$ rows if $A$ has at most $d \operatorname{polylog}(1/\epsilon)/\epsilon^2$ rows with leverage score at least $\epsilon/d$. This is a quadratic improvement in $d$ and an exponential improvement in $\delta$. In practice, it is not necessary to calculate the leverage scores. Instead, we show in our experiments that the indices of the rows of heavy leverage score can be learned and the induced $S$ is still accurate. We also consider a new learning objective, that is, we directly optimize the subspace embedding property of the sketching matrix instead of optimizing the error in *the objective function of the optimization problem* in hand. This demonstrates a significant advantage over non-learned sketches, and has a fast training time (Section 6).

We show strong empirical results for real-world datasets. For low-rank approximation, our methods reduce the errors by 70% than classical sketches under the same sketch size, while we reduce the errors by 30% than previous learning-based sketches. For second-order optimization, we show that the convergence rate can be reduced by 87% over the non-learned CountSketch for the LASSO problem on a real-world dataset. We also evaluate our approaches in the *few-shot learning* setting where we only have a limited amount of training data (Indyk et al., 2021). We show our approach reduces the error significantly even if we only have *one* training matrix (Sections 7 and 8). This approach clearly runs faster than all previous methods.

**Additional Related Work.**   In the last few years, there has been much work on leveraging machine learning techniques to improve classical algorithms. We only mention a few examples here which are based on learned sketches. One related body of work is data-dependent dimensionality reduction, such as an approach for pair-wise/multi-wise similarity preservation for indexing big data (Wang et al., 2017), learned sketching for streaming problems (Indyk et al., 2019; Aamand et al., 2019; Jiang et al., 2020; Cohen et al., 2020; Eden et al., 2021; Indyk et al., 2021), learned algorithms for nearest neighbor search (Dong et al., 2020), and a method for learning linear projections for general applications (Hegde et al., 2015). While we also learn linear embeddings, our embeddings are optimized for the specific application of low rank approximation. In fact, one of our central challenges is that the theory and practice of learned sketches generally needs to be tailored to each application. Our work builds off of (Indyk et al., 2019), which introduced gradient descent optimization for LRA, but a major difference is that we also optimize the locations of the non-zero entries.

## 2   PRELIMINARIES

**Notation.**   Denote the canonical basis vectors of $\mathbb{R}^n$ by $e_1, \ldots, e_n$. Suppose that $A$ has singular value decomposition (SVD) $A = U\Sigma V^\top$. Define $[A]_k = U_k\Sigma_k V_k^\top$ to be the optimal rank-$k$ approximation to $A$, computed by the truncated SVD. Also, define the Moore-Penrose pseudo-inverse of $A$ to be $A^\dagger = V\Sigma^{-1}U^\top$, where $\Sigma^{-1}$ is constructed by inverting the non-zero diagonal entries. Let $\operatorname{row}(A)$ and $\operatorname{col}(A)$ be the row space and the column space of $A$, respectively.

**CountSketch.**   We define $S_C \in \mathbb{R}^{m \times n}$ as a classical CountSketch (abbreviated as CS). It is a sparse matrix with one nonzero entry from $\{\pm 1\}$ per column. The position and value of this nonzero entry are chosen uniformly at random. CountSketch matrices can be succinctly represented by two vectors. We define $p \in [m]^n, v \in \mathbb{R}^n$ as the positions and values of the nonzero entries, respectively. Further, we let $\mathsf{CS}(p, v)$ be the CountSketch constructed from vectors $p$ and $v$.

Below we define the objective function $f(\cdot, \cdot)$ and a classical sketching algorithm $\mathsf{ALG}(\mathcal{S}, A)$ for each individual problem.

**Low-rank approximation (LRA).** In LRA, we find a rank-$k$ approximation of our data that minimizes the Frobenius norm of the approximation error. For $A \in \mathbb{R}^{n \times d}$, $\min_{\text{rank-}k\ B} f_{\text{LRA}}(A, B) = \min_{\text{rank-}k\ X} \|A - B\|_F^2$. Usually, instead of outputting the a whole $B \in \mathbb{R}^{n \times d}$, the algorithm outputs two factors $Y \in \mathbb{R}^{n \times k}$ and $X \in \mathbb{R}^{k \times d}$ such that $B = YX$ for efficiency.

Indyk et al. (2019) considered Algorithm 1, which only compresses one side of the input matrix $A$. However, in practice often both dimensions of the matrix $A$ are large. Hence, in this work we consider Algorithm 2 that compresses both sides of $A$.

**Constrained regression.** Given a vector $b \in \mathbb{R}^n$, a matrix $A \in \mathbb{R}^{n \times d}$ ($n \gg d$) and a convex set $\mathcal{C}$, we want to find $x$ to minimize the squared error

$$\min_{x \in \mathcal{C}} f_{\text{REG}}([A \ b], X) = \min_{x \in \mathcal{C}} \|Ax - b\|_2^2. \tag{2.1}$$

**Iterative Hessian Sketch.** The Iterative Hessian Sketching (IHS) method (Pilanci & Wainwright, 2016) solves the constrained least-squares problem by iteratively performing the update

$$x_{t+1} = \underset{x \in \mathcal{C}}{\arg\min} \left\{ \frac{1}{2} \|S_{t+1}A(x - x_t)\|_2^2 - \langle A^\top(b - Ax_t), x - x_t \rangle \right\}, \tag{2.2}$$

where $S_{t+1}$ is a sketching matrix. It is not difficult to see that for the unsketched version ($S_{t+1}$ is the identity matrix) of (2.2), the optimal solution $x^{t+1}$ coincides with the optimal solution to the original constrained regression problem (2.1). The IHS approximates the Hessian $A^\top A$ by a sketched version $(S_{t+1}A)^\top(S_{t+1}A)$ to improve runtime, as $S_{t+1}A$ typically has very few rows.

---

**Algorithm 1** Rank-$k$ approximation of $A$ using a sketch $S$ (see (Clarkson & Woodruff, 2009, Sec. 4.1.1))

---

**Input:** $\mathbf{A} \in \mathbb{R}^{n \times d}, S \in \mathbb{R}^{m \times n}$
1: $U, \Sigma, V^\top \leftarrow \text{COMPACTSVD}(SA)$ ▷ $\{r = \text{rank}(SA), U \in \mathbb{R}^{m \times r}, V \in \mathbb{R}^{d \times r}\}$
2: **Return:** $[AV]_k V^\top$

---

**Algorithm 2** $\text{ALG}_{\text{LRA}}(\text{SKETCH-LOWRANK})$ Sarlos (2006); Clarkson & Woodruff (2017); Avron et al. (2017).

---

**Input:** $A \in \mathbb{R}^{n \times d}, S \in \mathbb{R}^{m_S \times n}, R \in \mathbb{R}^{m_R \times d}$, $V \in \mathbb{R}^{m_V \times n}, W \in \mathbb{R}^{m_W \times d}$

1: $U_C [T_C \ T_C'] \leftarrow VAR^\top, \begin{bmatrix} T_D^\top \\ T_D'^\top \end{bmatrix} U_D^\top \leftarrow SAW^\top$ with $U_C, U_D$ orthogonal
2: $G \leftarrow VAW^\top, \ Z_L' Z_R' \leftarrow [U_C^\top G U_D]_k$
3: $Z_L \leftarrow [Z_L'(T_D^{-1})^\top \quad 0], Z_R \leftarrow \begin{bmatrix} T_C^{-1} Z_R' \\ 0 \end{bmatrix}$
4: $Z \leftarrow Z_L Z_R$
5: **return:** $AR^\top ZSA$ in form $P_{n \times k}, Q_{k \times d}$

---

**Learning-Based Algorithms in the Few-Shot Setting.** Recently, Indyk et al. (2021) studied learning-based algorithms for LRA in the setting where we have access to limited data or computing resources. We provide a brief explanation of learning-based algorithms in the Few-Shot setting in Appendix A.3.

**Leverage Scores and Ridge Leverage Scores.** Given a matrix $A$, the leverage score of the $i$-th row $a_i$ of $A$ is defined to be $\tau_i := a_i(A^\top A)^\dagger a_i^\top$, which is the squared $\ell_2$-norm of the $i$-th row of $U$, where $A = U \Sigma V^T$ is the singular value decomposition of $A$. Given a regularization parameter $\lambda$, the ridge leverage score of the $i$-th row $a_i$ of $A$ is defined to be $\tau_i := a_i(A^\top A + \lambda I)^\dagger a_i^\top$. Our learning-based algorithms employs the ridge leverage score sampling technique proposed in (Cohen et al., 2017), which shows that sampling proportional to ridge leverage scores gives a good solution to LRA.

## 3 DESCRIPTION OF OUR APPROACH

We describe our contributions to the learning-based sketching paradigm which, as mentioned, is to learn *the locations of the non-zero values* in the sketch matrix. To learn a CountSketch for the given training data set, we locally optimize the following in two stages:

$$\min_S \mathbb{E}_{A \in \mathcal{D}} [f(A, \text{ALG}(S, A))]. \tag{3.1}$$

(1) compute the positions of the non-zero entries, then (2) fix the positions and optimize their values.

**Stage 1: Optimizing Positions.** In Section 4, we provide a greedy search algorithm for this stage, as our starting point. In Section 5 and 6, we provide our specific approaches for optimizing the positions for the sketches for low-rank approximation and second-order optimization.

**Stage 2: Optimizing Values.** This stage is similar to the approach of Indyk et al. (2019). However, instead of the power method, we use an automatic differentiation package, PyTorch (Paszke et al., 2019), and we pass it our objective $\min_{v \in \mathbb{R}^n} \mathbb{E}_{A \in \mathcal{D}} [f(A, \text{ALG}(\text{CS}(p, v), A))]$, implemented as a chain of differentiable operations. It will automatically compute the gradient using the chain rule. We

also consider new approaches to optimize the values for LRA (proposed in Indyk et al. (2021), see Appendix A.3 for details) and second-order optimization (proposed in Section 6).

**Worst-Cases Guarantees.** In Appendix D, we show that both of our approaches for the above two problems can perform no worse than a classical sketching matrix when $A$ does not follow the distribution $\mathcal{D}$. In particular, for LRA, we show that the sketch monotonicity property holds for the time-optimal sketching algorithm for low rank approximation. For second-order optimization, we propose an algorithm which runs in input-sparsity time and can test for and use the better of a random sketch and a learned sketch.

## 4 SKETCH LEARNING: GREEDY SEARCH

When $S$ is a CountSketch, computing $SA$ amounts to hashing the $n$ rows of $A$ into the $m \ll n$ rows of $SA$. The optimization is a combinatorial optimization problem with an empirical risk minimization (ERM) objective. The naïve solution is to compute the objective value of the exponentially many ($m^n$) possible placements, but this is clearly intractable. Instead, we iteratively construct a full placement in a greedy fashion. We start with $S$ as a zero matrix. Then, we iterate through the columns of $S$ in an order determined by the algorithm, adding a nonzero entry to each. The best position in each column

---

**Algorithm 3** POSITION OPTIMIZATION: GREEDY SEARCH

**Input:** $f, \mathsf{ALG}, \mathsf{Tr} = \{A_1, ..., A_N \in \mathbb{R}^{n \times d}\}$; sketch dimension $m$
1: **initialize** $S_L = \mathbb{O}^{m \times n}$
2: **for** $i = 1$ to $n$ **do**
3:    $\bar{j} \leftarrow \underset{j \in [m]}{\arg\min} \sum_{A \in \mathsf{Tr}} f(A, \mathsf{ALG}(S_L \pm e_j e_i^\top, A))$
4:    $S_L \leftarrow S_L \pm (e_{\bar{j}} e_i^\top)$
5: **end for**
6: **return** $p$ for $S_L = \mathsf{CS}(p, v)$

---

is the one that minimizes Eq. (3.1) if an entry were to be added there. For each column, we evaluate Eq. (3.1) $\mathcal{O}(m)$ times, once for each prospective half-built sketch.

While this greedy strategy is simple to state, additional tactics are required for each problem to make it more tractable. Usually the objective evaluation (Algorithm 3, line 3) is too slow, so we must leverage our insight into their sketching algorithms to pick a proxy objective. Note that we can reuse these proxies for value optimization, since they may make gradient computation faster too.

**Proxy objective for LRA.** For the two-sided sketching algorithm, we can assume that the two factors $X, Y$ has the form $Y = AR^\top \tilde{Y}$ and $X = \tilde{X}SA$, where $S$ and $R$ are both CS matrices, so we optimize the positions in both $S$ and $R$. We cannot use $f(A, \mathsf{ALG}(S, R, A))$ as our objective because then we would have to consider *combinations* of placements between $S$ and $R$. To find a proxy, we note that a prerequisite for good performance is for $\mathsf{row}(SA)$ and $\mathsf{col}(AR^\top)$ to both contain a good rank-$k$ approximation to $A$ (see proof of Lemma C.5). Thus, we can decouple the optimization of $S$ and $R$. The proxy objective for $S$ is $\left\| [AV]_k V^\top - A \right\|_F^2$ where $SA = U\Sigma V^\top$. In this expression, $\hat{X} = [AV]_k V^\top$ is the best rank-$k$ approximation to $A$ in $\mathsf{row}(SA)$. The proxy objective for $R$ is defined analogously.

In Appendix F, we show the greedy algorithm is provably beneficial for LRA when inputs follow the spiked covariance or the Zipfian distribution. Despite the good empirical performance we present in Section 7, one drawback is its much slower training time. Also, for the iterative sketching method for second-order optimization, it is non-trivial to find a proxy objective because the input of the $i$-th iteration depends on the solution to the $(i-1)$-th iteration, for which the greedy approach sometimes does not give a good solution. In the next section, we will propose our specific approach for optimizing the positions of the sketches for low-rank approximation and second-order optimization, both of which achieve a very high accuracy and can finish in a very short amount of time.

## 5 SKETCH LEARNING: LOW-RANK APPROXIMATION

Now we present a conceptually new algorithm which runs much faster and empirically achieves similar error bounds as the greedy search approach. Moreover, we show that this algorithm has strictly better guarantees than the classical Count-Sketch.

To achieve this, we need a more careful analysis. To provide some intuition, if $\mathrm{rank}(SA) = k$ and $SA = U\Sigma V^\top$, then the rank-$k$ approximation cost is exactly $\left\| AVV^\top - A \right\|_F^2$, the projection cost

onto $\mathrm{col}(V)$. Minimizing it is equivalent to maximizing the sum of squared projection coefficients:

$$\arg\min_S \left\| A - AVV^\top \right\|_F^2 = \arg\min_S \sum_{i\in[n]} (\|A_i\|_2^2 - \sum_{j\in[k]} \langle A_i, v_j\rangle^2) = \arg\max_S \sum_{i\in[n]}\sum_{j\in[k]} \langle A_i, v_j\rangle^2.$$

As mentioned, computing $SA$ actually amounts to hashing the $n$ rows of $A$ to the $m$ rows of $SA$. Hence, intuitively, if we can put similar rows into the same bucket, we may get a smaller error.

---

**Algorithm 4** Position optimization: Inner Product

**Input:** $A \in \mathbb{R}^{n\times d}$: average of Tr; sketch dim. $m$
1: **initialize** $S_1, S_2 = \mathbb{0}^{m\times n}$
2: Sample a set $C = \{C_1 \cdots C_m\}$ of rows using ridge leverage score sampling (see Section 2).
3: **for** $i = 1$ to $n$ **do**
4: $\quad p_i, v_i \leftarrow \arg\max_{p\in[m], v\in\{\pm 1\}} \left\langle \frac{C_p}{\|C_p\|_2}, v\frac{A_i}{\|A_i\|_2} \right\rangle$
5: $\quad S_1[p_i, i] \leftarrow v_i$
6: **end for**
7: **for** $i = 1$ to $m$ **do**
8: $\quad I_i \leftarrow \{j \mid p_j = i\}$
9: $\quad A^{(i)} \leftarrow$ restriction of $A$ to rows in $I_i$
10: $\quad u_i \leftarrow$ the top left singular vector of $A^{(i)}$
11: $\quad S_1[i, I_i] \leftarrow u_i^\top$
12: **end for**
13: **for** $i = 1$ to $m$ **do**
14: $\quad q_i \leftarrow$ index such that $C_i$ is the $q_i$-th row of $A$
15: $\quad S_2[i, q_i] \leftarrow 1$
16: **end for**
17: **return** $S_1$ or $\begin{bmatrix} S_1 \\ S_2 \end{bmatrix}$

---

Our algorithm is given in Algorithm 4. Suppose that we want to form matrix $S$ with $m$ rows. At the beginning of the algorithm, we sample $m$ rows according to the ridge leverage scores of $A$. By the property of the ridge leverage score, the subspace spanned by this set of sampled rows contains an approximately optimal solution to the low rank approximation problem. Hence, we map these rows to separate "buckets" of $SA$. Then, we need to decide the locations of the remaining rows (i.e., the non-sampled rows). Ideally, we want similar rows to be mapped into the same bucket. To achieve this, we use the $m$ sampled rows as reference points and assign each (non-sampled) row $A_i$ to the $p$-th bucket in $SA$ if the normalized row $A_i$ and $C_p$ have the largest inner product (among all possible buckets).

Once the locations of the non-zero entries are fixed, the next step is to determine the values of these entries. We follow the same idea proposed in (Indyk et al., 2021): for each block $A^{(i)}$, one natural approach is to choose the unit vector $s_i \in \mathbb{R}^{|I_i|}$ that preserves as much of the Frobenius norm of $A^{(i)}$ as possible, i.e., to maximize $\left\| s_i^\top A^{(i)} \right\|_2^2$. Hence, we set $s_i$ to be the top left singular vector of $A^{(i)}$. In our experiments, we observe that this step reduces the error of downstream value optimizations performed by SGD.

To obtain a worst-case guarantee, we show that w.h.p., the row span of the sampled rows $C_i$ is a good subspace. We set the matrix $S_2$ to be the sampling matrix that samples $C_i$. The final output of our algorithm is the vertical concatenation of $S_1$ and $S_2$. Here $S_1$ performs well empirically, while $S_2$ has a worst-case guarantee for any input. Combining Lemma E.2 and the sketch monotonicity for low rank approximation in Section D, we get that $O(k\log k + k/\epsilon)$ rows is enough for a $(1\pm\epsilon)$-approximation for the input matrix $A$ induced by Tr, which is better than the $\Omega(k^2)$ rows required of a non-learned Count-Sketch, even if its non-zero values have been further improved by the previous learning-based algorithms in (Indyk et al., 2019; 2021). As a result, under the assumption of the input data, we may expect that $S$ will still be good for the test data. We defer the proof to Appendix E.1.

In Appendix A, we shall show that the assumptions we make in Theorem 5.1 are reasonable. We also provide an empirical comparison between Algorithm 4 and some of its variants, as well as some adaptive sketching methods on the training sample. The evaluation result shows that only our algorithm has a significant improvement for the test data, which suggests that both ridge leverage score sampling and row bucketing are essential.

**Theorem 5.1.** *Let $S \in \mathbb{R}^{2m\times n}$ be given by concatenating the sketching matrices $S_1, S_2$ computed by Algorithm 4 with input $A$ induced by Tr and let $B \in \mathbb{R}^{n\times d}$. Then with probability at least $1 - \delta$, we have $\min_{\mathrm{rank}\text{-}k\ X:\mathrm{row}(X)\subseteq\mathrm{row}(SB)} \|B - X\|_F^2 \leq (1+\epsilon)\|B - B_k\|_F^2$ if one of the following holds:.*

1. *$m = O(\beta\cdot(k\log k + k/\epsilon))$, $\delta = 0.1$, and $\tau_i(B) \geq \frac{1}{\beta}\tau_i(A)$ for all $i \in [n]$.*
2. *$m = O(k\log k + k/\epsilon)$, $\delta = 0.1 + 1.1\beta$, and the total variation distance $d_{\mathrm{tv}}(p, q) \leq \beta$, where $p, q$ are sampling probabilities defined as $p_i = \frac{\tau_i(A)}{\sum_i \tau_i(A)}$ and $q_i = \frac{\tau_i(B)}{\sum_i \tau_i(B)}$.*

**Time Complexity.** As mentioned, an advantage of our second approach is that it significantly reduces the training time. We now discuss the training times of different algorithms. For the value-learning

algorithms in (Indyk et al., 2019), each iteration requires computing a differentiable SVD to perform gradient descent, hence the runtime is at least $\Omega(n_{it} \cdot T)$, where $n_{it}$ is the number of iterations (usually set $> 500$) and $T$ is the time to compute an SVD. For the greedy algorithm, there are $m$ choices for each column, hence the runtime is at least $\Omega(mn \cdot T)$. For our second approach, the most complicated step is to compute the ridge leverage scores of $A$ and then the SVD of each submatrix. Hence, the total runtime is at most $O(T)$. We note that the time complexities discussed here are all for training time. There is no additional runtime cost for the test data.

## 6 SKETCH LEARNING: SECOND-ORDER OPTIMIZATION

In this section, we consider optimizing the sketch matrix in the context of second-order methods. The key observation is that for many sketching-based second-order methods, the crucial property of the sketching matrix is the so-called subspace embedding property: for a matrix $A \in \mathbb{R}^{n \times d}$, we say a matrix $S \in \mathbb{R}^{m \times n}$ is a $(1 \pm \epsilon)$-subspace embedding for the column space of $A$ if $(1 - \epsilon) \|Ax\|_2 \leq \|SAx\|_2 \leq (1 + \epsilon) \|Ax\|_2$ for all $x \in \mathbb{R}^d$. For example, consider the iterative Hessian sketch, which performs the update (2.2) to compute $\{x_t\}_t$. Pilanci & Wainwright (2016) showed that if $S_1, \ldots, S_{t+1}$ are $(1 + O(\rho))$-subspace embeddings of $A$, then $\|A(x^t - x^*)\|_2 \leq \rho^t \|Ax^*\|_2$. Thus, if $S_i$ is a good subspace embedding of $A$ and we will have a good convergence guarantee. Therefore, unlike (Indyk et al., 2019), which treats the training objective in a black-box manner, we shall optimize the subspace embedding property of the matrix $A$.

**Optimizing positions.** We consider the case that $A$ has a few rows of large leverage score, as well as access to an oracle which reveals a *superset* of the indices of such rows. Formally, let $\tau_i(A)$ be the leverage score of the $i$-th row of $A$ and $I^* = \{i : \tau_i(A) \geq \nu\}$ be the set of rows with large leverage score. Suppose that a superset $I \supseteq I^*$ is known to the algorithm. In the experiments we train an oracle to predict such rows. We can maintain all rows in $I$ explicitly and apply a Count-Sketch to the remaining rows, i.e., the rows in $[n] \setminus I$. Up to permutation of the rows, we can write

$$A = \begin{pmatrix} A_I \\ A_{I^c} \end{pmatrix} \quad \text{and} \quad S = \begin{pmatrix} I & 0 \\ 0 & S' \end{pmatrix}, \tag{6.1}$$

where $S'$ is a random Count-Sketch matrix of $m$ rows. Clearly $S$ has a single non-zero entry per column. We have the following theorem, whose proof is postponed to Section E.2. Intuitively, the proof for Count-Sketch in (Clarkson & Woodruff, 2017) handles rows of large leverage score and rows of small leverage score separately. The rows of large leverage score are to be perfectly hashed while the rows of small leverage score will concentrate in the sketch by the Hanson-Wright inequality.

**Theorem 6.1.** *Let $\nu = \epsilon/d$. Suppose that $m = O((d/\epsilon^2)(\mathrm{polylog}(1/\epsilon) + \log(1/\delta)))$, $\delta \in (0, 1/m]$ and $d = \Omega((1/\epsilon) \mathrm{polylog}(1/\epsilon) \log^2(1/\delta))$. Then, there exists a distribution on $S$ of the form in (6.1) with $m + |I|$ rows such that $\Pr \left\{ \forall x \in \mathrm{col}(A), | \|Sx\|_2^2 - \|x\|_2^2 | > \epsilon \|x\|_2^2 \right\} \leq \delta$. In particular, when $\delta = 1/m$, the sketching matrix $S$ has $O((d/\epsilon^2) \mathrm{polylog}(d/\epsilon))$ rows.*

Hence, if there happen to be at most $d \, \mathrm{polylog}(1/\epsilon)/\epsilon^2$ rows of leverage score at least $\epsilon/d$, the overall sketch length for embedding $\mathrm{colsp}(A)$ can be reduced to $O((d \, \mathrm{polylog}(1/\epsilon) + \log(1/\delta))/\epsilon^2)$, a quadratic improvement in $d$ and an exponential improvement in $\delta$ over the original sketch length of $O(d^2/(\epsilon^2 \delta))$ for Count-Sketch. In the worst case there could be $O(d^2/\epsilon)$ such rows, though empirically we do not observe this. In Section 8, we shall show it is possible to learn the indices of the heavy rows for real-world data.

**Optimizing values.** When we fix the positions of the non-zero entries, we aim to optimize the values by gradient descent. Rather than the previous black-box way in (Indyk et al., 2019) that minimizes $\sum_i f(A, \mathsf{ALG}(S, A))$, we propose the following objective loss function for the learning algorithm $\mathcal{L}(S, \mathcal{A}) = \sum_{A_i \in \mathcal{A}} \|(A_i R_i)^\top A_i R_i - I\|_F$, over all the training data, where $R_i$ comes from the QR decomposition of $SA_i = Q_i R_i^{-1}$. The intuition for this loss function is given by the lemma below, whose proof is deferred to Section E.3.

**Lemma 6.2.** *Suppose that $\epsilon \in (0, \frac{1}{2})$, $S \in \mathbb{R}^{m \times n}$, $A \in \mathbb{R}^{n \times d}$ of full column rank, and $SA = QR$ is the QR-decomposition of $SA$. If $\|(AR^{-1})^\top AR^{-1} - I\|_{\mathrm{op}} \leq \epsilon$, then $S$ is a $(1 \pm \epsilon)$-subspace embedding of $\mathrm{col}(A)$.*

Lemma 6.2 implies that if the loss function over $\mathcal{A}_{\text{train}}$ is small and the distribution of $\mathcal{A}_{\text{test}}$ is similar to $\mathcal{A}_{\text{train}}$, it is reasonable to expect that $S$ is a good subspace embedding of $\mathcal{A}_{\text{test}}$. Here we use the Frobenius norm rather than operator norm in the loss function because it will make the optimization problem easier to solve, and our empirical results also show that the performance of the Frobenius norm is better than that of the operator norm.

# 7 EXPERIMENTS: LOW-RANK APPROXIMATION

In this section, we evaluate the empirical performance of our learning-based approach for LRA on three datasets. For each, we fix the sketch size and compare the approximation error $\|A - X\|_F - \|A - A_k\|_F$ averaged over 10 trials. In order to make position optimization more efficient, in line 3 of Algorithm 3), instead of computing many rank-1 SVD updates, we use formulas for *fast* rank-1 SVD updates (Brand, 2006). For the greedy method, we used several Nvidia GeForce GTX 1080 Ti machines. For the maximum inner product method, the experiments are conducted on a laptop with a 1.90GHz CPU and 16GB RAM.

**Datasets.** We use the three datasets from (Indyk et al., 2019): (1, 2) **Friends, Logo** (image): frames from a short video of the TV show *Friends* and of a logo being painted; (3) **Hyper** (image): hyperspectral images from natural scenes. Additional details are in Table A.1.

**Baselines.** We compare our approach to the following baselines. **Classical CS**: a random Count-Sketch. **IVY19**: a sparse sketch with learned values, and random positions for the non-zero entries. **Ours (greedy)**: a sparse sketch where both the values and positions of the non-zero entries are learned. The positions are learned by Algorithm 3. The values are learned similarly to (Indyk et al., 2019). **Ours (inner product)**: a sparse sketch where both the values and the positions of the non-zero entries are learned. The positions are learned by $S_1$ in Algorithm 4. IVY19 and greedy algorithm use the full training set and our Algorithm 4 takes the input as the average over the entire training matrix.

We also give a sensitivity analysis for our algorithm, where we compare our algorithm with the following variants: **Only row sampling** (perform projection by ridge leverage score sampling), $\ell_2$ **sampling** (Replace leverage score sampling with $\ell_2$-norm row sampling and maintain the same downstream step), and **Randomly Grouping** (Use ridge leverage score sampling but randomly distribute the remaining rows). The result shows none of these variants outperforms non-learned sketching. We defer the results of this part to Appendix A.1.

**Result Summary.** Our empirical results are provided in Table 7.1 for both Algorithm 2 and Algorithm 1, where the errors take an average over 10 trials. We use the average of all training matrices from Tr, as the input to the algorithm 4. We note that all the steps of our training algorithms are done on the training data. Hence, no additional computational cost is incurred for the sketching algorithm on the test data. Experimental parameters (i.e., learning rate for gradient descent) can be found in Appendix G. For both sketching algorithms, **Ours** are always the best of the four sketches. It is significantly better than **Classical CS**, obtaining improvements of around 70%. It also obtains a roughly 30% improvement over **IVY19**.

| | Offline learning | Online solving |
|---|---|---|
| Ours (inner product) | 5 | 0.166 |
| Ours (greedy) | 6300 (1.75h) | 0.172 |
| IVY19 | 193 (3min) | 0.168 |
| Classical CS | ✗ | 0.166 |

Table 7.2: Runtime (in seconds) of LRA on Logo with $k = 30, m = 60$

**Wall-Clock Times.** The offline learning runtime is in Table 7.2, which is the time to train a sketch on $\mathcal{A}_{\text{train}}$. We can see that although the greedy method will take much longer (1h 45min), our second approach is much faster (5 seconds) than the previous algorithm in (Indyk et al., 2019) (3 min) and can still achieve a similar error as the greedy

| $k, m$, Sketch | Logo | Friends | Hyper | $k, m$, Sketch | Logo | Friends | Hyper |
|---|---|---|---|---|---|---|---|
| 20, 40, Classical CS | 2.371 | 4.073 | 6.344 | 20, 40, Classical CS | 0.930 | 1.542 | 2.971 |
| 20, 40, IVY19 | 0.687 | 1.048 | 3.764 | 20, 40, IVY19 | 0.255 | 0.723 | 1.273 |
| 20, 40, Ours (greedy) | **0.500** | 0.899 | **2.497** | 20, 40, Ours (greedy) | **0.196** | **0.407** | **0.784** |
| 20, 40, Ours (inner product) | 0.532 | **0.733** | 2.975 | 20, 40, Ours (inner product) | 0.205 | **0.407** | 1.223 |
| 30, 60, Class CS | 1.642 | 2.683 | 5.390 | 30, 60, Classical CS | 0.650 | 1.0575 | 2.315 |
| 30, 60, IVY19 | 0.734 | 1.077 | 3.748 | 30, 60, IVY19 | 0.290 | 0.713 | 1.274 |
| 30, 60, Ours (greedy) | 0.492 | 0.794 | 2.492 | 30, 60, Ours(greedy) | **0.197** | 0.406 | **0.717** |
| 30, 60, Ours (inner product) | **0.436** | **0.733** | **2.409** | 30, 60, Ours(inner product) | 0.201 | **0.340** | 0.943 |

Table 7.1: Test errors for LRA. (Left: two-side sketch. Right: one-side sketch)

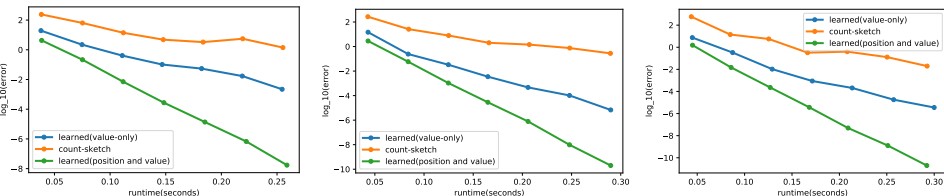

Figure 7.1: Test error of LASSO in Electric dataset.

algorithm. The reason is that Algorithm 4 only needs to compute the ridge leverage scores on the training matrix once, which is actually much cheaper than **IVY19** which needs to compute a differentiable SVD many times during gradient descent.

In Section A.4, we also study the performance of our approach in the few-shot learning setting, which has been studied in Indyk et al. (2021).

## 8 EXPERIMENTS: SECOND-ORDER OPTIMIZATION

In this section, we consider the IHS on the following instance of LASSO regression:

$$x^* = \arg\min_{\|x\|_1 \le \lambda} f(x) = \arg\min_{\|x\|_1 \le \lambda} \tfrac{1}{2} \|Ax - b\|_2^2, \tag{8.1}$$

where $\lambda$ is a parameter. We also study the performance of the sketches on the matrix estimation with a nuclear norm constraint problem, the fast regression solver (van den Brand et al. (2021)), as well as the use of sketches for first-order methods. The results can be found in Appendix B. All of our experiments are conducted on a laptop with a 1.90GHz CPU and 16GB RAM. The offline training is done separately using a single GPU. The details of the implementation are deferred to Appendix G.

**Dataset.** We use the Electric[1] dataset of residential electric load measurements. Each row of the matrix corresponds to a different residence. Matrix columns are consecutive measurements at different times. Here $A^i \in \mathbb{R}^{370 \times 9}$, $b^i \in \mathbb{R}^{370 \times 1}$, and $|(A, b)_{\text{train}}| = 320$, $|(A, b)_{\text{test}}| = 80$. We set $\lambda = 15$.

**Experiment Setting.** We compare the learned sketch against the classical Count-Sketch[2]. We choose $m = 6d, 8d, 10d$ and consider the error $f(x) - f(x^*)$. For the heavy-row Count-Sketch, we allocate 30% of the sketch space to the rows of the heavy row candidates. For this dataset, each row represents a specific residence and hence there is a strong pattern of the distribution of the heavy rows. We select the heavy rows according to the number of times each row is heavy in the training data. We give a detailed discussion about this in Appendix B.1. We highlight that it is still possible to recognize the pattern of the rows even if the row orders of the test data are permuted. We also consider optimizing the non-zero values after identifying the heavy rows, using our new approach in Section 6.

**Results.** We plot in Figures 7.1 the mean errors on a logarithmic scale. The average offline training time is 3.67s to find a superset of the heavy rows over the training data and 66s to optimize the values when $m = 10d$, which are both faster than the runtime of Indyk et al. (2019) with the same parameters. Note that the learned matrix $S$ is trained offline only once using the training data. *Hence, no additional computational cost is incurred when solving the optimization problem on the test data.*

We see all methods display linear convergence, that is, letting $e_k$ denote the error in the $k$-th iteration, we have $e_k \approx \rho^k e_1$ for some convergence rate $\rho$. A smaller convergence rate implies a faster convergence. We calculate an estimated rate of convergence $\rho = (e_k/e_1)^{1/k}$ with $k = 7$. We can see both sketches, especially the sketch that optimizes both the positions and values, show significant improvements. When the sketch size is small ($6d$), this sketch has a convergence rate that is just 13.2% of that of the classical Count-Sketch, and when the sketch size is large ($10d$), this sketch has a smaller convergence rate that is just 12.1%.

---

[1]https://archive.ics.uci.edu/ml/datasets/ElectricityLoadDiagrams20112014
[2]The framework of Indyk et al. (2019) does not apply to the iterative sketching methods in a straightforward manner, so here we only compare with the classical CountSketch. For more details, please refer to Section B.

ACKNOWLEDGEMENTS

Yi Li would like to thank for the partial support from the Singapore Ministry of Education under Tier 1 grant RG75/21. Honghao Lin and David Woodruff were supported in part by an Office of Naval Research (ONR) grant N00014-18-1-2562. Ali Vakilian was supported by NSF award CCF-1934843.

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

## A ADDITIONAL EXPERIMENTS: LOW-RANK APPROXIMATION

The details (data dimension, $N_{\text{train}}$, etc.) are presented in Table A.1.

| Name | Type | Dimension | $N_{\text{train}}$ | $N_{\text{test}}$ |
|---|---|---|---|---|
| Friends | Image | $5760 \times 1080$ | 400 | 100 |
| Logo | Image | $5760 \times 1080$ | 400 | 100 |
| Hyper | Image | $1024 \times 768$ | 400 | 100 |

Table A.1: Data set descriptions

### A.1 SENSITIVITY ANALYSIS OF ALGORITHM 4

| Sketch | Logo | Friends | Hyper |
|---|---|---|---|
| Ours(inner product) | **0.311** | **0.470** | **1.232** |
| $\ell_2$ Sampling | 0.698 | 0.935 | 1.293 |
| Only Ridge | 0.994 | 1.493 | 4.155 |
| Randomly Grouping | 0.659 | 1.069 | 2.070 |

Table A.2: Sensitivity analysis for our approach (using Algorithm 1 from Indyk et al. (2019) with one sketch)

In this section we explore how sensitive the performance of our Algorithm 4 is to the ridge leverage score sampling and maximum inner product grouping process. We consider the following baselines:

- $\ell_2$ norm sampling: we sample the rows according to their squared length instead of doing ridge leverage score sampling.

- Only ridge leverage score sampling: the subspace spanned by only the sampled rows from ridge leverage score sampling.

- Randomly grouping: we put the sampled rows into different buckets as before, but randomly divide the non-sampled rows into buckets.

The results are shown in Table A.2. Here we set $k = 30, m = 60$ as an example. To show the difference of the initialization method more clearly, we compare the error using the one-sided sketching Algorithm 1 and do not further optimize the non-zeros values. From the table we can see both that ridge leverage score sampling and the downstream grouping process are necessary, otherwise the error will be similar or even worse than that of the classical Count-Sketch.

### A.2 TOTAL VARIATION DISTANCE

As we have shown in Theorem 5.1, if the total variation distance between the row sampling probability distributions $p$ and $q$ is $O(1)$, we have a worst-case guarantee of $O(k \log k + k/\epsilon)$, which is strictly better than the $\Omega(k^2)$ lower bound for the random CountSketch, even when its non-zero values have been optimized. We now study the total variation distance between the train and test matrix in our dataset. The result is shown in Figure A.1. From the figure we can see that for all the three dataset, the total variation distance is bounded by a constant, which suggests that the assumptions are reasonable for real-world data.

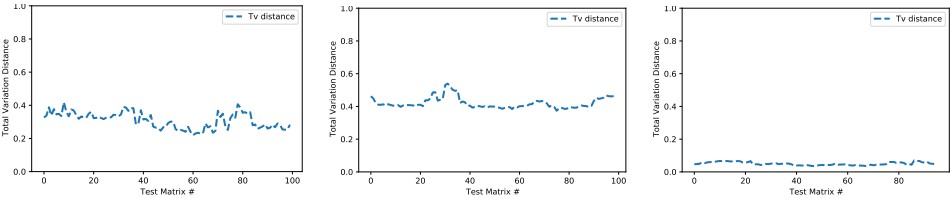

Figure A.1: Total variation distance between train and test matrices. left: Logo, middle: friend, right: Hyper.

### A.3 Learning-Based Algorithms for Low-Rank Approximation in the Few-Shot Setting

In this section, we will give a brief explanation of the two algorithms proposed in Indyk et al. (2021). Both algorithms aim to optimize the non-zero values of a Count-Sketch matrix under fixed locations of the non-zero entries.

**One-shot closed-form algorithm.** Given a sparsity pattern of a Count-Sketch matrix $S \in \mathbb{R}^{m \times n}$, it partitions the rows of $A$ into $m$ blocks $A^{(1)}, ..., A^{(m)}$ as follows: let $I_i = \{j : S_{ij} = 1\}$. The block $A^{(i)} \in \mathbb{R}^{|I_i| \times d}$ is the sub-matrix of $A$ that contains the rows whose indices are in $I_i$. The goal here is for each block $A^{(i)}$, to choose a (non-sparse) one-dimensional sketching vector $s_i \in \mathbb{R}^{|I_i|}$. The first approach is to set $s_i$ to be the top left singular vector of $A^{(i)}$, which is the algorithm **1Shot2Vec**. Another approach is to set $s_i$ to be a left singular vector of $A^{(i)}$ chosen randomly and proportional to its squared singular value. The main advantage of the latter approach over the previous one is that it endows the algorithm with provable guarantees on the LRA error. The **1Shot2Vec** algorithm combines both ways, obtaining the benefits of both approaches. The advantage of these two algorithms is that they extract a sketching matrix by an analytic computation, requiring neither GPU access nor auto-gradient functionality.

**Few-shot SGD algorithm.** In this algorithm, the authors propose a new loss function for LRA, namely,

$$\min_{\mathsf{CS}} \mathop{\mathbf{E}}_{S} \mathop{}_{A \in \mathsf{Tr}} \left\| U_k^\top S^\top S U - I_0 \right\|_F^2 \ ,$$

where $A = U\Sigma V^\top$ is the SVD-decomposition of $A$ and $U_k \in \mathbb{R}^{n \times k}$ denotes the submatrix of $U$ that contains its first $k$ columns. $I_0 \in \mathbb{R}^{k \times d}$ denotes the result of augmenting the identity matrix of order $k$ with $d - k$ additional zero columns on the right. This loss function is motivated by the analysis of prior LRA algorithms that use random sketching matrices. It is faster to compute and differentiate than the previous empirical loss in Indyk et al. (2019). In the experiments the authors also show that this loss function can achieve a smaller error in a shorter amount of time, using a small number of randomly sampled training matrices, though the final error will be larger than that of the previous algorithm in Indyk et al. (2019) if we allow a longer training time and access to the whole training set Tr.

### A.4 Experiments: LRA in the few-shot setting

In the rest of this section, we study the performance of our second approach in the few-shot learning setting. We first consider the case where we only have one training matrix randomly sampled from Tr. Here, we compare our method with the **1Shot2Vec** method proposed in (Indyk et al., 2021) in the same setting ($k = 10, m = 40$) as in their empirical evaluation. The result is shown in Table A.3. Compared to **1Shot2Vec**, our method reduces the error by around $50\%$, and has an even slightly faster runtime.

Indyk et al. (2021) also proposed a **FewShotSGD** algorithm which further improves the non-zero values of the sketches after different initialization methods. We compare the performance of this approach for different initialization methods: in all initialization methods, we only use one training matrix and we use three training matrices for the **FewShotSGD** step. The results are shown in Table A.4. We report the minimum error of 50 iterations of the **FewShotSGD** because we aim to compare the computational efficiency for different methods. From the table we see that our approach plus the **FewShotSGD** method can achieve a much smaller error, with around a $50\%$ improvement upon (Indyk et al., 2021). Moreover, even without further optimization by **FewShotSGD**, our initialization method for learning the non-zero locations in CountSketch obtains a smaller error than other methods (even when they are optimized with **1ShotSGD** or **FewShotSGD** learning).

## B Additional Experiments: Second-Order Optimization

As we mentioned in Section 8, despite the number of problems that learned sketches have been applied to, they have not been applied to convex optimization, or say, iterative sketching algorithms in general. To demonstrate the difficulty, we consider the Iterative Hessian Sketch (IHS) as an example. In that scheme, suppose that we have $k$ iterations of the algorithm. Then we need $k$ independent

| Algorithm | Dataset | Few-shot Error | Training Time |
|---|---|---|---|
| Classical CS | Logo | 0.331 | ✗ |
| | Friends | 0.524 | |
| | Hyper | 1.082 | |
| 1shot2Vec | Logo | 0.171 | 5.682 |
| | Friends | 0.306 | 5.680 |
| | Hyper | 0.795 | 1.054 |
| Ours (inner product) | Logo | 0.065 | 4.515 |
| | Friends | 0.139 | 4.773 |
| | Hyper | 0.535 | 0.623 |

Table A.3: Test errors and training times for LRA in the one-shot setting (using Alg. 1 with one sketch)

| Sketch | Logo | Friends | Hyper |
|---|---|---|---|
| Ours (Initialization only) | 0.065 | 0.139 | 0.535 |
| Ours + FewShotSGD | **0.048** | **0.125** | **0.443** |
| 1Shot1Vec only | 0.171 | 0.306 | 0.795 |
| 1Shot1Vec + FewShot SGD | 0.104 | 0.229 | 0.636 |
| Classical CS | 0.331 | 0.524 | 1.082 |
| Classical CS + FewShot SGD | 0.173 | 0.279 | 0.771 |

Table A.4: Test errors for LRA in the few-shot setting (using Alg. 1 from Indyk et al. (2019) with one sketch)

sketching matrices (otherwise the solution may diverge). A natural way is to follow the method in (Indyk et al., 2019), which is to minimize the following quantity

$$\min_{S_1,...S_k} \mathop{\mathbf{E}}_{A \in \mathcal{D}} f(A, \mathsf{ALG}(S_1, ..., S_k, A)) \,,$$

where the minimization is taken over $k$ Count-Sketch matrices $S_1, \ldots, S_k$. In this case, however, calculating the gradient with respect to $S_1$ would involve all iterations and in each iteration we need to solve a constrained optimization problem. Hence, it would be difficult and intractable to compute the gradients. An alternative way is to train $k$ sketching matrices sequentially, that is, learn the sketching matrix for the $i$-th iteration using a local loss function for the $i$-th iteration, and then using the learned matrix in the $i$-th iteration to generate the training data for the $(i + 1)$-st iteration. However, the empirical results suggest that it works for the first iteration only, because in this case the training data for the $(i + 1)$-st iteration depends on the solution to the $i$-th iteration and may become farther away from the test data in later iterations. The core problem here is that the method proposed in Indyk et al. (2019) treats the training process in a *black-box* way, which is difficult to extend to iterative methods.

## B.1 THE DISTRIBUTION OF THE HEAVY ROWS

In our experiments, we hypothesize that in real-world data there may be an underlying pattern which can help us identify the heavy rows. In the Electric dataset, each row of the matrix corresponds to a specific residence and the heavy rows are concentrated on some specific rows.

To exemplify this, we study the heavy leverage score rows distribution over the Electric dataset. For a row $i \in [370]$, let $f_i$ denote the number of times that row $i$ is heavy out of 320 training data points from the Electric dataset, where we say row $i$ is heavy if $\ell_i \geq 5d/n$. Below we list all 74 pairs $(i, f_i)$ with $f_i > 0$.

(195,320), (278,320), (361,320), (207,317), (227,285), (240,284), (219,270), (275,232), (156,214), (322,213), (193,196), (190,192), (160,191), (350,181), (63,176), (42,168), (162,148), (356,129), (363,110), (362,105), (338,95), (215,94), (234,93), (289,81), (97,80), (146,70), (102,67), (98,58), (48,57), (349,53), (165,46), (101,41), (352,40), (293,34), (344,29), (268,21), (206,20), (217,20), (327,20), (340,19), (230,18), (359,18), (297,14), (357,14), (161,13), (245,10), (100,8), (85,6), (212,6), (313,6), (129,5), (130,5), (366,5), (103,4), (204,4), (246,4), (306,4), (138,3), (199,3), (222,3), (360,3), (87,2), (154,2), (209,2), (123,1), (189,1), (208,1), (214,1), (221,1), (224,1), (228,1), (309,1), (337,1), (343,1)

Observe that the heavy rows are concentrated on a set of specific row indices. There are only 30 rows $i$ with $f_i \geq 50$. We view this as strong evidence for our hypothesis.

**Heavy Rows Distribution Under Permutation.** We note that even though the order of the rows has been changed, we can still recognize the patterns of the rows. We continue to use the Electric dataset as an example. To address the concern that a permutation may break the sketch, we can measure the similarity between vectors, that is, after processing the training data, we can instead test similarity on the rows of the test matrix and use this to select the heavy rows, rather than an index which may simply be permuted. To illustrate this method, we use the following example on the Electric dataset, using locality sensitive hashing.

After processing the training data, we obtain a set $I$ of indices of heavy rows. For each $i \in I$, we pick $q = 3$ independent standard Gaussian vectors $g_1, g_2, g_3 \in \mathbb{R}^d$, and compute $f(r_i) =$

$(g_1^T r_i, g_2^T r_i, g_3^T r_i) \in \mathbb{R}^3$, where $r_i$ takes an average of the $i$-th rows over all training sets. Let $A$ be the test matrix. For each $i \in I$, let $j_i = \arg\min_j \|f(A_j) - f(r_i)\|_2$. We take the $j_i$-th row to be a heavy row in our learned sketch. This method only needs an additional $O(1)$ passes over the entries of $A$ and hence, the extra time cost is negligible. To test the performance of the method, we randomly pick a matrix from the test set and permute its rows. The result shows that when $k$ is small, we can roughly recover 70% of the top-$k$ heavy rows, and we plot below the regression error using the learned Count-Sketch matrix generated this way, where we set $m = 90$ and $k = 0.3m = 27$. We can see that the learned method still obtains a significant improvement.

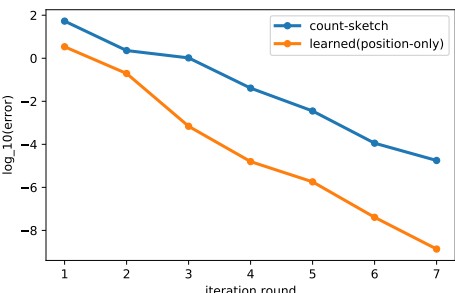

Figure B.1: Test error of LASSO on Electric dataset

## B.2 MATRIX NORM ESTIMATION WITH A NUCLEAR NORM CONSTRAINT

In many applications, for the problem

$$X^* := \arg\min_{X \in \mathbb{R}^{d_1 \times d_2}} \|AX - B\|_F^2 \ ,$$

it is reasonable to model the matrix $X^*$ as having low rank. Similar to $\ell_1$-minimization for compressive sensing, a standard relaxation of the rank constraint is to minimize the nuclear norm of $X$, defined as $\|X\|_* := \sum_{j=1}^{\min\{d_1, d_2\}} \sigma_j(X)$, where $\sigma_j(X)$ is the $j$-th largest singular value of $X$.

Hence, the matrix estimation problem we consider here is

$$X^* := \arg\min_{X \in \mathbb{R}^{d_1 \times d_2}} \|AX - B\|_F^2 \quad \text{such that} \quad \|X\|_* \leq \rho,$$

where $\rho > 0$ is a user-defined radius as a regularization parameter.

We conduct Iterative Hessian Sketch (IHS) experiments on the following dataset:

- **Tunnel**[3]: The data set is a time series of gas concentrations measured by eight sensors in a wind tunnel. Each $(A, B)$ corresponds to a different data collection trial. $A^i \in \mathbb{R}^{13530 \times 5}, B^i \in \mathbb{R}^{13530 \times 6}, |(A, B)|_{\text{train}} = 144, |(A, B)|_{\text{test}} = 36$. In our nuclear norm constraint, we set $\rho = 10$.

**Experiment Setting**: We choose $m = 7d, 10d$ for the Tunnel dataset. We consider the error $\frac{1}{2}\|AX - B\|_2^2 - \frac{1}{2}\|AX^* - B\|_2^2$. The leverage scores of this dataset are very uniform. Hence, for this experiment we only consider optimizing the values of the non-zero entries.

**Results of Our Experiments**: We plot on a logarithmic scale the mean errors of the dataset in Figures B.2. We can see that when $m = 7d$, the gradient-based sketch, based on the first 6 iterations, has a rate of convergence that is 48% of the random sketch, and when $m = 10d$, the gradient-based sketch has a rate of convergence that is 29% of the random sketch.

## B.3 FAST REGRESSION SOLVER

Consider an unconstrained convex optimization problem $\min_x f(x)$, where $f$ is smooth and strongly convex, and its Hessian $\nabla^2 f$ is Lipschitz continuous. This problem can be solved by Newton's

---

[3] https://archive.ics.uci.edu/ml/datasets/Gas+sensor+array+exposed+to+turbulent+gas+mixtures

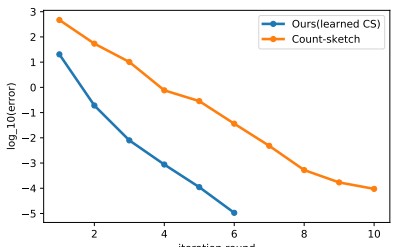 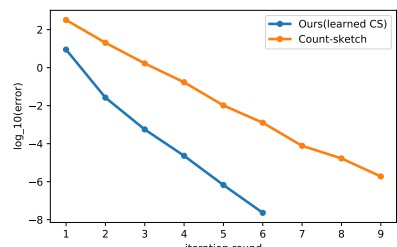

Figure B.2: Test error of matrix estimation with a nuclear norm constraint on the Tunnel dataset

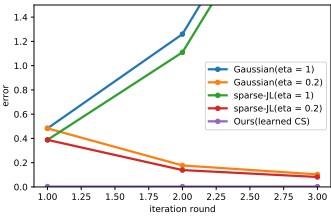 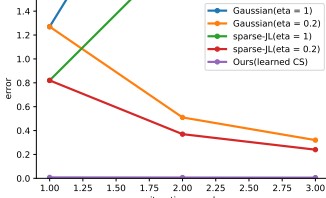 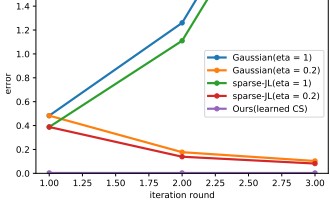

Figure B.3: Test error of the subroutine in fast regression on Electric dataset.

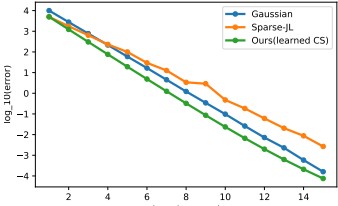

Figure B.4: Test error of fast regression on Electric dataset

method, which iteratively performs the update

$$x_{t+1} = x_t - \arg\min_z \left\| (\nabla^2 f(x_t)^{1/2})^\top (\nabla^2 f(x_t)^{1/2}) z - \nabla f(x_t) \right\|_2, \tag{B.1}$$

provided it is given a good initial point $x_0$. In each step, it requires solving a regression problem of the form $\min_z \left\| A^\top A z - y \right\|_2$, which, with access to $A$, can be solved with a fast regression solver in (van den Brand et al., 2021). The regression solver first computes a preconditioner $R$ via a QR decomposition such that $SAR$ has orthonormal columns, where $S$ is a sketching matrix, then solves $\hat{z} = \arg\min_{z'} \left\| (AR)^\top (AR) z' - y \right\|_2$ by gradient descent and returns $R\hat{z}$ in the end. Here, the point of sketching is that the QR decomposition of $SA$ can be computed much more efficiently than the QR decomposition of $A$, since $S$ has only a small number of rows.

In this section, We consider the unconstrained least squares problem $\min_x f(x)$ with $f(x) = \frac{1}{2} \left\| Ax - b \right\|_2^2$ using the Electric dataset, using the above fast regression solver.

**Training**: Note that $\nabla^2 f(x) = A^\top A$, independent of $x$. In the $t$-th round of Newton's method, by (B.1), we need to solve a regression problem $\min_z \left\| A^\top A z - y \right\|_2^2$ with $y = \nabla f(x_t)$. Hence, we can use the same methods in the preceding subsection to optimize the learned sketch $S_i$. For a general problem where $\nabla^2 f(x)$ depends on $x$, one can take $x_t$ to be the solution obtained from the learned sketch $S_t$ to generate $A$ and $y$ for the $(t+1)$-st round, train a learned sketch $S_{t+1}$, and repeat this process.

**Experiment Setting**: For the Electric dataset, we set $m = 10d = 90$. We observe that the classical Count-Sketch matrix makes the solution diverge terribly in this setting. To make a clearer comparison, we consider the following sketch matrix:

- Gaussian sketch: $S = \frac{1}{\sqrt{m}} G$, where $G \in \mathbb{R}^{m \times n}$ with i.i.d. $N(0, 1)$ entries.

- Sparse Johnson-Lindenstrauss Transform (SJLT): $S$ is the vertical concatenation of $s$ independent Count-Sketch matrices, each of dimension $m/s \times n$.

We note that the above sketching matrices require more time to compute $SA$ but need fewer rows to be a subspace embedding than the classical Count-Sketch matrix.

For the step length $\eta$ in gradient descent, we set $\eta = 1$ in all iterations of the learned sketches. For classical random sketches, we set $\eta$ in the following two ways: (a) $\eta = 1$ in all iterations and (b) $\eta = 1$ in the first iteration and $\eta = 0.2$ in all subsequent iterations.

**Experimental Results**: We examine the accuracy of the subproblem $\min_z \left\| A^\top Az - y \right\|_2^2$ and define the error to be $\left\| A^\top ARz_t - y \right\|_2 / \left\| y \right\|_2$. We consider the subproblems in the first three iterations of the global Newton method. The results are plotted in Figure B.3. Note that Count-Sketch causes a terrible divergence of the subroutine and is thus omitted in the plots. Still, we observe that in setting (a) of $\eta$, the other two classical sketches cause the subroutine to diverge. In setting (b) of $\eta$, the other two classical sketches lead to convergence but their error is significantly larger than that of the learned sketches, in each of the first three calls to the subroutine. The error of the learned sketch is less than $0.01$ in all iterations of all three subroutine calls, in both settings (a) and (b) of $\eta$.

We also plot a figure on the convergence of the global Newton method. Here, for each subroutine, we only run one iteration, and plot the error of the original least squares problem. The result is shown in Figure B.4, which clearly displays a significantly faster decay with learned sketches. The rate of convergence using heavy-rows sketches is $80.6\%$ of that using Gaussian or sparse JL sketches.

### B.4 FIRST-ORDER OPTIMIZATION

In this section, we study the use of the sketch in first-order methods. Particularly, let $QR^{-1} = SA$ be the QR-decomposition for $SA$, where $S$ is a sketch matrix. We use $R$ as an (approximate) preconditioner and use gradient descent to solve the problem $\min \left\| ARx - b \right\|_2$. Here we use the Electric dataset where $A$ is $370 \times 9$ and we set $S$ to have $90$ rows. The result is shown in the following table, where the time includes the time to compute $R$. We can see that if we use a learned sketch matrix, the error converges very fast when we set the learning rate to be $1$ and $0.1$, while the classical Count-Sketch will lead to divergence.

| Iteration | 1 | 10 | 100 | 500 |
|---|---|---|---|---|
| Error (learned, $lr = 1$) | 2.73 | 1.5e-7 | | |
| Error (learned, $lr = 0.1$) | 4056 | 605 | 4.04e-6 | |
| Error (learned, $lr = 0.01$) | 4897 | 4085 | 667 | 0.217 |
| Error (random, $lr = 1$) | N.A | N.A | N.A | N.A |
| Error (random, $lr = 0.1$) | | | | |
| Error (random, $lr = 0.01$) | 4881 | 3790 | 685 | 1.52 |
| Time | 0.00048 | 0.00068 | 0.0029 | 0.0013 |

Table B.1: Test Error for Gradient Descent

## C PRELIMINARIES: THEOREMS AND ADDITIONAL ALGORITHMS

In this section, we provide the full description of the time-optimal sketching algorithm for LRA in Algorithm 2. We also provide several definitions and lemmas that are used in the proofs of our results for LRA.

**Definition C.1** (Affine Embedding). Given a pair of matrices $A$ and $B$, a matrix $S$ is an *affine $\epsilon$-embedding* if for all $X$ of the appropriate shape, $\left\| S(AX - B) \right\|_F^2 = (1 \pm \epsilon) \left\| AX - B \right\|_F^2$.

**Lemma C.2** (Clarkson & Woodruff (2017); Lemma 40). *Let $A$ be an $n \times d$ matrix and let $S \in \mathbb{R}^{\mathcal{O}(1/\epsilon^2) \times n}$ be a CountSketch matrix. Then with constant probability, $\left\| SA \right\|_F^2 = (1 \pm \epsilon) \left\| A \right\|_F^2$.*

The following result is shown in Clarkson & Woodruff (2017) and sharpened with Nelson & Nguyên (2013); Meng & Mahoney (2013).

**Lemma C.3.** *Given matrices $A, B$ with $n$ rows, a CountSketch with $\mathcal{O}(\mathrm{rank}(A)^2/\epsilon^2)$ rows is an affine $\epsilon$-embedding matrix with constant probability. Moreover, the matrix product $SA$ can be computed in $\mathcal{O}(\mathrm{nnz}(A))$ time, where $\mathrm{nnz}(A)$ denotes the number of non-zero entries of matrix $A$.*

**Lemma C.4** (Sarlos (2006); Clarkson & Woodruff (2017))**.** *Suppose that $A \in \mathbb{R}^{n \times d}$ and $B \in \mathbb{R}^{n \times d'}$. Let $S \in \mathbb{R}^{m \times n}$ be a CountSketch with $m = \frac{\mathrm{rank}(A)^2}{\epsilon^2}$. Let $\tilde{X} = \arg\min_{\mathrm{rank}\text{-}k\ X} \|SAX - SB\|_F^2$. Then,*

1. *With constant probability, $\left\| A\tilde{X} - B \right\|_F^2 \le (1 + \epsilon) \min_{\mathrm{rank}\text{-}k\ X} \|AX - B\|_F^2$. In other words, in $\mathcal{O}(\mathrm{nnz}(A) + \mathrm{nnz}(B) + m(d + d'))$ time, we can reduce the problem to a smaller (multi-response regression) problem with $m$ rows whose optimal solution is a $(1 + \epsilon)$-approximate solution to the original instance.*

2. *The $(1 + \epsilon)$-approximate solution $\tilde{X}$ can be computed in time $\mathcal{O}(\mathrm{nnz}(A) + \mathrm{nnz}(B) + mdd' + \min(m^2 d, md^2))$.*

Now we turn our attention to the time-optimal sketching algorithm for LRA. The next lemma is known, though we include it for completeness Avron et al. (2017):

**Lemma C.5.** *Suppose that $S \in \mathbb{R}^{m_S \times n}$ and $R \in \mathbb{R}^{m_R \times d}$ are sparse affine $\epsilon$-embedding matrices for $(A_k, A)$ and $((SA)^\top, A^\top)$, respectively. Then,*

$$\min_{\mathrm{rank}\text{-}k\ X} \left\| AR^\top XSA - A \right\|_F^2 \le (1 + \epsilon) \|A_k - A\|_F^2$$

*Proof.* Consider the following multiple-response regression problem:

$$\min_{\mathrm{rank}\text{-}k\ X} \|A_k X - A\|_F^2 . \tag{C.1}$$

Note that since $X = I_k$ is a feasible solution to Eq. (C.1), $\min_{\mathrm{rank}\text{-}k\ X} \|A_k X - A\|_F^2 = \|A_k - A\|_F^2$. Let $S \in \mathbb{R}^{m_S \times n}$ be a sketching matrix that satisfies the condition of Lemma C.4 (Item 1) for $A := A_k$ and $B := A$. By the normal equations, the rank-$k$ minimizer of $\|SA_k X - SA\|_F^2$ is $(SA_k)^+ SA$. Hence,

$$\left\| A_k (SA_k)^+ SA - A \right\|_F^2 \le (1 + \epsilon) \|A_k - A\|_F^2 , \tag{C.2}$$

which in particular shows that a $(1 + \epsilon)$-approximate rank-$k$ approximation of $A$ exists in the row space of $SA$. In other words,

$$\min_{\mathrm{rank}\text{-}k\ X} \|XSA - A\|_F^2 \le (1 + \epsilon) \|A_k - A\|_F^2 . \tag{C.3}$$

Next, let $R \in \mathbb{R}^{m_R \times d}$ be a sketching matrix which satisfies the condition of Lemma C.4 (Item 1) for $A := (SA)^\top$ and $B := A^\top$. Let $Y$ denote the rank-$k$ minimizer of $\left\| R(SA)^\top X^\top - RA^\top \right\|_F^2$. Hence,

$$\left\| (SA)^\top Y^\top - A^\top \right\|_F^2 \le (1 + \epsilon) \min_{\mathrm{rank}\text{-}k\ X} \|XSA - A\|_F^2 \qquad \triangleright \text{Lemma C.4 (Item 1)}$$

$$\le (1 + \mathcal{O}(\epsilon)) \|A_k - A\|_F^2 \qquad \triangleright \text{Eq. (C.3)} \tag{C.4}$$

Note that by the normal equations, again $\mathrm{rowsp}(Y^\top) \subseteq \mathrm{rowsp}(RA^\top)$ and we can write $Y = AR^\top Z$ where $\mathrm{rank}(Z) = k$. Thus,

$$\min_{\mathrm{rank}\text{-}k\ X} \left\| AR^\top XSA - A \right\|_F^2 \le \left\| AR^\top ZSA - A \right\|_F^2 = \left\| (SA)^\top Y^\top - A^\top \right\|_F^2 \qquad \triangleright Y = AR^\top Z$$

$$\le (1 + \mathcal{O}(\epsilon)) \|A_k - A\|_F^2 \qquad \triangleright \text{Eq. (C.4)}$$

$\square$

**Lemma C.6** (Avron et al. (2017); Lemma 27)**.** *For $C \in \mathbb{R}^{p \times m'}, D \in \mathbb{R}^{m \times p'}, G \in \mathbb{R}^{p \times p'}$, the following problem*

$$\min_{\mathrm{rank}\text{-}k\ Z} \|CZD - G\|_F^2 \tag{C.5}$$

*can be solved in $\mathcal{O}(pm' r_C + p' m r_D + pp'(r_D + r_C))$ time, where $r_C = \mathrm{rank}(C) \le \min\{m', p\}$ and $r_D = \mathrm{rank}(D) \le \min\{m, p'\}$.*

**Lemma C.7.** *Let $S \in \mathbb{R}^{m_S \times d}$, $R \in \mathbb{R}^{m_R \times d}$ be CountSketch (CS) matrices such that*

$$\min_{\text{rank-}k \ X} \left\| AR^\top XSA - A \right\|_F^2 \leq (1 + \gamma) \left\| A_k - A \right\|_F^2 . \tag{C.6}$$

*Let $V \in \mathbb{R}^{\frac{m_R^2}{\beta^2} \times n}$, and $W \in \mathbb{R}^{\frac{m_S^2}{\beta^2} \times d}$ be CS matrices. Then, Algorithm 2 gives a $(1 + \mathcal{O}(\beta + \gamma))$-approximation in time $\mathrm{nnz}(A) + \mathcal{O}(\frac{m_S^4}{\beta^2} + \frac{m_R^4}{\beta^2} + \frac{m_S^2 m_R^2(m_S + m_R)}{\beta^4} + k(nm_S + dm_R))$ with constant probability.*

*Proof.* The approximation guarantee follows from Eq. (C.6) and the fact that $V$ and $W$ are affine $\beta$-embedding matrices of $AR^\top$ and $SA$, respectively (see Lemma C.3).

The algorithm first computes $C = VAR^\top, D = SAW^\top, G = VAW^\top$ which can be done in time $\mathcal{O}(\mathrm{nnz}(A))$. As an example, we bound the time to compute $C = VAR$. Note that since $V$ is a CS, $VA$ can be computed in $\mathcal{O}(\mathrm{nnz}(A))$ time and the number of non-zero entries in the resulting matrix is at most $\mathrm{nnz}(A)$. Hence, since $R$ is a CS as well, $C$ can be computed in time $\mathcal{O}(\mathrm{nnz}(A) + \mathrm{nnz}(VA)) = \mathcal{O}(\mathrm{nnz}(A))$. Then, it takes an extra $\mathcal{O}((m_S^3 + m_R^3 + m_S^2 m_R^2)/\beta^2)$ time to store $C, D$ and $G$ in matrix form. Next, as we showed in Lemma C.6, the time to compute $Z$ in Algorithm 2 is $\mathcal{O}(\frac{m_S^4}{\beta^2} + \frac{m_R^4}{\beta^2} + \frac{m_S^2 m_R^2(m_S + m_R)}{\beta^4})$. Finally, it takes $\mathcal{O}(\mathrm{nnz}(A) + k(nm_S + dm_R))$ time to compute $Q = AR^\top Z_L$ and $P = Z_R SA$ and to return the solution in the form of $P_{n \times k} Q_{k \times d}$. Hence, the total runtime is

$$\mathcal{O}(\mathrm{nnz}(A) + \frac{m_S^4}{\beta^2} + \frac{m_R^4}{\beta^2} + \frac{m_S^2 m_R^2(m_S + m_R)}{\beta^4} + k(nm_S + dm_R))$$

$\square$

# D   ATTAINING WORST-CASE GUARANTEES

## D.1   LOW-RANK APPROXIMATION

We shall provide the following two methods to achieve worst case guarantees: MixedSketch—whose guarantee is via the sketch monotonicity property, and approximate comparison method (a.k.a. ApproxCheck), which just approximately evaluates the cost of two solutions and takes the better one. These methods asymptotically achieve the same worst-case guarantee. However, for any input matrix $A$ and any pair of sketches $S, T$, the performance of the MixedSketch method on $(A, S, T)$ is never worse than the performance of its corresponding ApproxCheck method on $(A, S, T)$, and can be much better.

*Remark D.1.* Let $A = \mathrm{diag}(2, 2, \sqrt{2}, \sqrt{2})$, and suppose the goal is to find a rank-2 approximation of $A$. Consider two sketches $S$ and $T$ such that $SA$ and $TA$ capture $\mathsf{span}(e_1, e_3)$ and $\mathsf{span}(e_2, e_4)$, respectively. Then for both $SA$ and $TA$, the best solution in the subspace of one of these two spaces is a $(\frac{3}{2})$-approximation: $\|A - A_2\|_F^2 = 4$ and $\|A - P_{SA}\|_F^2 = \|A - P_{TA}\|_F^2 = 6$ where $P_{SA}$ and $P_{TA}$ respectively denote the best approximation of $A$ in the space spanned by $SA$ and $TA$.

However, if we find the best rank-2 approximation of $A$, $Z$, inside the span of the union of $SA$ and $TA$, then $\|A - Z\|_F^2 = 4$. Since ApproxCheck just chooses the better of $SA$ and $TA$ by evaluating their costs, it misses out on the opportunity to do as well as MixedSketch.

Here, we show the sketch monotonicity property for LRA.

**Theorem D.2.** *Let $A \in \mathbb{R}^{n \times d}$ be an input matrix, $V$ and $W$ be $\eta$-affine embeddings, and $S_1 \in \mathbb{R}^{m_S \times n}, R_1 \in \mathbb{R}^{m_R \times n}$ be arbitrary matrices. Consider arbitrary extensions to $S_1, R_1$: $\overline{S}, \overline{R}$ (e.g., $\overline{S}$ is a concatenation of $S_1$ with an arbitrary matrix with the same number of columns). Then, $\left\| A - \mathsf{ALG}_{\mathrm{LRA}}((\overline{S}, \overline{R}, V, W), A) \right\|_F^2 \leq (1 + \eta)^2 \left\| A - \mathsf{ALG}_{\mathrm{LRA}}((S_1, R_1, V, W), A) \right\|_F^2$*

*Proof.* We have $\left\| A - \mathsf{ALG}_{\mathrm{LRA}}((\overline{S}, \overline{R}, V, W), A) \right\|_F^2 \leq (1 + \eta) \min_{\text{rank-}k \ X} \left\| A\overline{R} X \overline{S} A - A \right\|_F^2 = (1 + \eta) \min_{\text{rank-}k \ X : X \in \mathsf{row}(\overline{S}A) \cap \mathsf{col}(A\overline{R})} \|X - A\|_F^2$, which is in turn at most $(1 + \eta) \min_{\text{rank-}k \ X : X \in \mathsf{row}(S_1 A) \cap \mathsf{col}(AR_1)} \|X - A\|_F^2 = (1 + \eta) \min_{\text{rank-}k \ X} \left\| AR_1 X S_1 A - A \right\|_F^2 \leq (1 +$

$\eta)^2 \|A - \mathsf{ALG}_{\mathrm{LRA}}((S_1, R_1, V, W), A)\|_F^2$, where we use the fact the $V, W$ are affine $\eta$-embeddings (Definition C.1), as well as the fact that $(\mathsf{col}(AR_1) \cap \mathsf{row}(S_1 A)) \subseteq (\mathsf{col}(A\overline{R}) \cap \mathsf{row}(\overline{S}A))$. $\qquad\square$

**ApproxCheck for LRA.** We give the pseudocode for the ApproxCheck method and prove that the runtime of this method for LRA is of the same order as the classical time-optimal sketching algorithm of LRA.

---
**Algorithm 5** LRA APPROXCHECK
---
**Input:** learned sketches $S_L, R_L, V_L, W_L$; classical sketches $S_C, R_C, V_C, W_C$; $\beta$; $A \in \mathbb{R}^{n \times d}$
1: $P_L, Q_L \leftarrow \mathsf{ALG}_{\mathrm{LRA}}(S_L, R_L, V_L, W_L, A)$, $P_C Q_C \leftarrow \mathsf{ALG}_{\mathrm{LRA}}(S_C, R_C, V_C, W_C, A)$
2: Let $S \in \mathbb{R}^{\mathcal{O}(1/\beta^2) \times n}, R \in \mathbb{R}^{\mathcal{O}(1/\beta^2) \times d}$ be classical CountSketch matrices
3: $\Delta_L \leftarrow \left\|S(P_L Q_L - A)R^\top\right\|_F^2, \Delta_C \leftarrow \left\|S(P_C Q_C - A)R^\top\right\|_F^2$
4: **if** $\Delta_L \leq \Delta_C$ **then**
5: $\quad$ **return** $P_L Q_L$
6: **end if**
7: **return** $P_C Q_C$

---

**Theorem D.3.** *Assume we have data $A \in \mathbb{R}^{n \times d}$, learned sketches $S_L \in \mathbb{R}^{\mathrm{poly}(\frac{k}{\epsilon}) \times n}, R_L \in \mathbb{R}^{\mathrm{poly}(\frac{k}{\epsilon}) \times d}, V_L \in \mathbb{R}^{\mathrm{poly}(\frac{k}{\epsilon}) \times n}, W_L \in \mathbb{R}^{\mathrm{poly}(\frac{k}{\epsilon}) \times d}$ which attain a $(1 + \mathcal{O}(\gamma))$-approximation, classical sketches of the same size, $S_C, R_C, V_C, W_C$, which attain a $(1 + \mathcal{O}(\epsilon))$-approximation, and a tradeoff parameter $\beta$. Then, Algorithm 5 attains a $(1 + \beta + \min(\gamma, \epsilon))$-approximation in $\mathcal{O}(\mathrm{nnz}(A) + (n + d) \mathrm{poly}(\frac{k}{\epsilon}) + \frac{k^4}{\beta^4} \cdot \mathrm{poly}(\frac{k}{\epsilon}))$ time.*

*Proof.* Let $(P_L, Q_L), (P_C, Q_C)$ be the approximate rank-$k$ approximations of $A$ in factored form using $(S_L, R_L)$ and $(S_O, R_O)$. Then, clearly,

$$\min(\|P_L Q_L - A\|_F^2, \|P_C Q_C - A\|_F^2) = (1 + \mathcal{O}(\min(\epsilon, \gamma))) \|A_k - A\|_F^2 \qquad (\mathrm{D.1})$$

Let $\Gamma_L = P_L Q_L - A, \Gamma_C = P_C Q_C - A$ and $\Gamma_M = \arg\min(\|S\Gamma_L R\|_F, \|S\Gamma_C R\|_F)$. Then,

$$\begin{aligned}
\|\Gamma_M\|_F^2 &\leq (1 + \mathcal{O}(\beta)) \|S\Gamma_M R\|_F^2 && \triangleright \text{ by Lemma C.2} \\
&\leq (1 + \mathcal{O}(\beta)) \cdot \min(\|\Gamma_L\|_F^2, \|\Gamma_C\|_F^2) \\
&\leq (1 + \mathcal{O}(\beta + \min(\epsilon, \gamma))) \|A_k - A\|_F^2 && \triangleright \text{ by Eq. (D.1)}
\end{aligned}$$

*Runtime analysis.* By Lemma C.7, Algorithm 2 computes $P_L, Q_L$ and $P_C, Q_C$ in time $\mathcal{O}(\mathrm{nnz}(A) + \frac{k^{16}(\beta^2 + \epsilon^2)}{\epsilon^{24}\beta^4} + \frac{k^3}{\epsilon^2}(n + \frac{dk^2}{\epsilon^4}))$. Next, once we have $P_L, Q_L$ and $P_C, Q_C$, it takes $\mathcal{O}(\mathrm{nnz}(A) + \frac{k}{\beta^4})$ time to compute $\Delta_L$ and $\Delta_C$.

$$\mathcal{O}\left(\mathrm{nnz}(A) + \frac{k^{16}(\beta^2 + \epsilon^2)}{\epsilon^{24}\beta^4} + \frac{k^3}{\epsilon^2}\left(n + \frac{dk^2}{\epsilon^4}\right) + \frac{k}{\beta^4}\right) = \mathcal{O}\left(\mathrm{nnz}(A) + \left(n + d + \frac{k^4}{\beta^4}\right)\mathrm{poly}\left(\frac{k}{\epsilon}\right)\right).$$

$\qquad\square$

To interpret the above theorem, note that when $\epsilon \gg k(n + d)^{-4}$, we can set $\beta^{-4} = \mathcal{O}(k(n + d)^{-4})$ so that Algorithm 5 has the same asymptotic runtime as the best $(1 + \epsilon)$-approximation algorithm for LRA with the classical CountSketch. Moreover, Algorithm 5 is a $(1 + o(\epsilon))$-approximation when the learned sketch outperforms classical sketches, $\gamma = o(\epsilon)$. On the other hand, when the learned sketches perform poorly, $\gamma = \Omega(\epsilon)$, the worst-case guarantee of Algorithm 5 remains $(1 + \mathcal{O}(\epsilon))$.

### D.2 SECOND-ORDER OPTIMIZATION

For the sketches for second-order optimization, the monotonicity property does not hold. Below we provide an input-sparsity algorithm which can test for and use the better of a random sketch and a learned sketch. Our theorem is as follows.

**Theorem D.4.** *Let $\epsilon \in (0, 0.09)$ be a constant and $S_1$ a learned Count-Sketch matrix. Suppose that $A$ is of full rank. There is an algorithm whose output is a solution $\hat{x}$ which, with probability at least $0.98$, satisfies that $\|A(\hat{x} - x^*)\|_2 \leq O\left(\min\left\{\frac{Z_2(S_1)}{Z_1(S_1)}, \epsilon\right\}\right) \|Ax^*\|_2$, where $x^* = \arg\min_{x \in \mathcal{C}} \|Ax - b\|_2$ is the least-squares solution. Furthermore, the algorithm runs in $O(\mathrm{nnz}(A) \log(\frac{1}{\epsilon}) + \mathrm{poly}(\frac{d}{\epsilon}))$ time.*

---

**Algorithm 6** Solver for (D.2)

---

1: $S_1 \leftarrow$ learned sketch, $S_2 \leftarrow$ random sketch with $\Theta(d^2/\epsilon^2)$ rows
2: $(\hat{Z}_{i,1}, \hat{Z}_{i,2}) \leftarrow \text{ESTIMATE}(S_i, A)$, $i = 1, 2$
3: $i^* \leftarrow \arg\min_{i=1,2}(\hat{Z}_{i,2}/\hat{Z}_{i,1})$
4: $\hat{x} \leftarrow$ solution of (D.2) with $S = S_{i^*}$
5: **return** $\hat{x}$

---

6: **function** ESTIMATE$(S, A)$
7: $T \leftarrow$ sparse $(1 \pm \eta)$-subspace embedding matrix for $d$-dimensional subspaces
8: $(Q, R) \leftarrow \text{QR}(TA)$
9: $\hat{Z}_1 \leftarrow \sigma_{\min}(SAR^{-1})$
10: $\hat{Z}_2 \leftarrow (1 \pm \eta)$-approximation to $\left\| (SAR^{-1})^\top (SAR^{-1}) - I \right\|_{\text{op}}$
11: **return** $(\hat{Z}_1, \hat{Z}_2)$

---

Consider the minimization problem

$$\min_{x \in \mathcal{C}} \left\{ \frac{1}{2} \left\| SAx \right\|_2^2 - \langle A^\top y, x \rangle \right\}, \tag{D.2}$$

which is used as a subroutine for the IHS (cf. (2.2)). We note that in this subroutine if we let $x \leftarrow x - x^{i-1}, b \leftarrow b - Ax^{i-1}, \mathcal{C} \leftarrow \mathcal{C} - x^{i-1}$, we would get the guarantee of the $i$-th iteration of the original IHS. To analyze the performance of the learned sketch, we define the following quantities (corresponding exactly to the unconstrained case in (Pilanci & Wainwright, 2016))

$$Z_1(S) = \inf_{v \in \text{colsp}(A) \cap \mathbb{S}^{n-1}} \left\| Sv \right\|_2^2,$$

$$Z_2(S) = \sup_{u, v \in \text{colsp}(A) \cap \mathbb{S}^{n-1}} \left\langle u, (S^\top S - I_n)v \right\rangle.$$

When $S$ is a $(1 + \epsilon)$-subspace embedding of $\text{colsp}(A)$, we have $Z_1(S) \geq 1 - \epsilon$ and $Z_2(S) \leq 2\epsilon$.

For a general sketching matrix $S$, the following is the approximation guarantee of $\hat{Z}_1$ and $\hat{Z}_2$, which are estimates of $Z_1(S)$ and $Z_2(S)$, respectively. The main idea is that $AR^{-1}$ is well-conditioned, where $R$ is as calculated in Algorithm 6.

**Lemma D.5.** *Suppose that $\eta \in (0, \frac{1}{3})$ is a small constant, $A$ is of full rank and $S$ has $\text{poly}(d/\eta)$ rows. The function ESTIMATE$(S, A)$ returns in $O((\text{nnz}(A) \log \frac{1}{\eta} + \text{poly}(\frac{d}{\eta}))$ time $\hat{Z}_1, \hat{Z}_2$ which with probability at least $0.99$ satisfy that $\frac{Z_1(S)}{1+\eta} \leq \hat{Z}_1 \leq \frac{Z_1(S)}{1-\eta}$ and $\frac{Z_2(S)}{(1+\eta)^2} - 3\eta \leq \hat{Z}_2 \leq \frac{Z_2(S)}{(1-\eta)^2} + 3\eta$.*

*Proof.* Suppose that $AR^{-1} = UW$, where $U \in \mathbb{R}^{n \times d}$ has orthonormal columns, which form an orthonormal basis of the column space of $A$. Since $T$ is a subspace embedding of the column space of $A$ with probability $0.99$, it holds for all $x \in \mathbb{R}^d$ that

$$\frac{1}{1+\eta} \left\| TAR^{-1}x \right\|_2 \leq \left\| AR^{-1}x \right\|_2 \leq \frac{1}{1-\eta} \left\| TAR^{-1}x \right\|_2.$$

Since

$$\left\| TAR^{-1}x \right\|_2 = \left\| Qx \right\|_2 = \left\| x \right\|_2$$

and

$$\left\| Wx \right\|_2 = \left\| UWx \right\|_2 = \left\| AR^{-1}x \right\|_2 \tag{D.3}$$

we have that

$$\frac{1}{1+\eta} \left\| x \right\|_2 \leq \left\| Wx \right\|_2 \leq \frac{1}{1-\eta} \left\| x \right\|_2, \quad x \in \mathbb{R}^d. \tag{D.4}$$

It is easy to see that

$$Z_1(S) = \min_{x \in \mathbb{S}^{d-1}} \left\| SUx \right\|_2 = \min_{y \neq 0} \frac{\left\| SUWy \right\|_2}{\left\| Wy \right\|_2},$$

and thus,

$$\min_{y \neq 0}(1 - \eta) \frac{\left\| SUWy \right\|_2}{\left\| y \right\|_2} \leq Z_1(S) \leq \min_{y \neq 0}(1 + \eta) \frac{\left\| SUWy \right\|_2}{\left\| y \right\|_2}.$$

Recall that $SUW = SAR^{-1}$. We see that

$$(1 - \eta)\sigma_{\min}(SAR^{-1}) \le Z_1(S) \le (1 + \eta)\sigma_{\min}(SAR^{-1}).$$

By definition,

$$Z_2(S) = \left\| U^T(S^\top S - I_n)U \right\|_{\mathrm{op}}.$$

It follows from (D.4) that

$$(1 - \eta)^2 \left\| W^T U^T(S^T S - I_n)UW \right\|_{\mathrm{op}} \le Z_2(S) \le (1 + \eta)^2 \left\| W^T U^T(S^T S - I_n)UW \right\|_{\mathrm{op}}.$$

and from (D.4), (D.3) and Lemma 5.36 of Vershynin (2012) that

$$\left\| (AR^{-1})^\top(AR^{-1}) - I \right\|_{\mathrm{op}} \le 3\eta.$$

Since

$$\left\| W^T U^T(S^T S - I_n)UW \right\|_{\mathrm{op}} = \left\| (AR^{-1})^\top(S^T S - I_n)AR^{-1} \right\|_{\mathrm{op}}$$

and

$$
\begin{aligned}
& \left\| (AR^{-1})^\top S^T SAR^{-1} - I \right\|_{\mathrm{op}} - \left\| (AR^{-1})^\top(AR^{-1}) - I \right\|_{\mathrm{op}} \\
&\le \left\| (AR^{-1})^\top(S^T S - I_n)AR^{-1} \right\|_{\mathrm{op}} \\
&\le \left\| (AR^{-1})^\top S^T SAR^{-1} - I \right\|_{\mathrm{op}} + \left\| (AR^{-1})^\top(AR^{-1}) - I \right\|_{\mathrm{op}},
\end{aligned}
$$

it follows that

$$
\begin{aligned}
& (1 - \eta)^2 \left\| (SAR^{-1})^\top SAR^{-1} - I \right\|_{\mathrm{op}} - 3(1 - \eta)^2\eta \\
&\le Z_2(S) \\
&\le (1 + \eta)^2 \left\| (SAR^{-1})^\top SAR^{-1} - I \right\|_{\mathrm{op}} + 3(1 + \eta)^2\eta.
\end{aligned}
$$

We have so far proved the correctness of the approximation and we next analyze the runtime below.

Since $S$ and $T$ are sparse, computing $SA$ and $TA$ takes $O(\mathrm{nnz}(A))$ time. The QR decomposition of $TA$, which is a matrix of size $\mathrm{poly}(d/\eta) \times d$, can be computed in $\mathrm{poly}(d/\eta)$ time. The matrix $SAR^{-1}$ can be computed in $\mathrm{poly}(d)$ time. Since it has size $\mathrm{poly}(d/\eta) \times d$, its smallest singular value can be computed in $\mathrm{poly}(d/\eta)$ time. To approximate $Z_2(S)$, we can use the power method to estimate $\left\| (SAR^{-1})^T SAR^{-1} - I \right\|_{op}$ up to a $(1 \pm \eta)$-factor in $O((\mathrm{nnz}(A) + \mathrm{poly}(d/\eta)) \log(1/\eta))$ time. $\square$

Now we are ready to prove Theorem D.4.

*Proof of Theorem D.4.* In Lemma D.5, we have with probability at least 0.99 that

$$\frac{\hat{Z}_2}{\hat{Z}_1} \ge \frac{\frac{1}{(1+\epsilon)^2} Z_2(S) - 3\epsilon}{\frac{1}{1-\epsilon} Z_1(S)} \ge \frac{1 - \epsilon}{(1 + \epsilon)^2} \frac{Z_2(S)}{Z_1(S)} - \frac{3\epsilon(1 - \epsilon)}{Z_1(S)}.$$

and similarly,

$$\frac{\hat{Z}_2}{\hat{Z}_1} \le \frac{\frac{1}{(1-\epsilon)^2} Z_2(S) + 3\epsilon}{\frac{1}{1+\epsilon} Z_1(S)} \le \frac{1 + \epsilon}{(1 - \epsilon)^2} \frac{Z_2(S)}{Z_1(S)} + \frac{3\epsilon(1 + \epsilon)}{Z_1(S)}.$$

Note that since $S_2$ is an $\epsilon$-subspace embedding with probability at least 0.99, we have that $Z_1(S_2) \ge 1 - \epsilon$ and $Z_2(S_2)/Z_1(S_2) \le 2.2\epsilon$. Consider $Z_1(S_1)$.

First, we consider the case where $Z_1(S_1) < 1/2$. Observe that $Z_2(S) \ge 1 - Z_1(S)$. We have in this case $\widehat{Z}_{1,2}/\widehat{Z}_{1,1} > 1/5 \ge 2.2\epsilon \ge Z_2(S_2)/Z_1(S_2)$. In this case our algorithm will choose $S_2$ correctly.

Next, assume that $Z_1(S_1) \ge 1/2$. Now we have with probability at least 0.98 that

$$(1 - 3\epsilon)\frac{Z_2(S_i)}{Z_1(S_i)} - 3\epsilon \le \frac{\widehat{Z}_{i,2}}{\widehat{Z}_{i,1}} \le (1 + 4\epsilon)\frac{Z_2(S_i)}{Z_1(S_i)} + 4\epsilon, \quad i = 1, 2.$$

Therefore, when $Z_2(S_1)/Z_1(S_1) \leq c_1 Z_2(S_2)/Z_1(S_2)$ for some small absolute constant $c_1 > 0$, we will have $\widehat{Z}_{1,2}/\widehat{Z}_{1,1} < \widehat{Z}_{2,2}/\widehat{Z}_{2,1}$, and our algorithm will choose $S_1$ correctly. If $Z_2(S_1)/Z_1(S_1) \geq C_1 \epsilon$ for some absolute constant $C_1 > 0$, our algorithm will choose $S_2$ correctly. In the remaining case, both ratios $Z_2(S_2)/Z_1(S_2)$ and $Z_2(S_1)/Z_1(S_1)$ are at most $\max\{C_2, 3\}\epsilon$, and the guarantee of the theorem holds automatically.

The correctness of our claim then follows from Proposition 1 of Pilanci & Wainwright (2016), together with the fact that $S_2$ is a random subspace embedding. The runtime follows from Lemma D.5 and Theorem 2.2 of Cormode & Dickens (2019). □

# E SKETCH LEARNING: OMITTED PROOFS

## E.1 PROOF OF THEOREM 5.1

We need the following lemmas for the ridge leverage score sampling in (Cohen et al., 2017).

**Lemma E.1** ((Cohen et al., 2017, Lemma 4)). *Let $\lambda = \|A - A_k\|_F^2 / k$. Then we have $\sum_i \tau_i(A) \leq 2k$.*

**Lemma E.2** ((Cohen et al., 2017, Theorem 7)). *Let $\lambda = \|A - A_k\|_F^2 / k$ and $\tilde{\tau}_i \geq \tau_i(A)$ be an overestimate to the $i$-th ridge leverage score of $A$. Let $p_i = \tilde{\tau}_i / \sum_i \tilde{\tau}_i$. If $C$ is a matrix that is constructed by sampling $t = O((\log k + \frac{\log(1/\delta)}{\epsilon}) \cdot \sum_i \tilde{\tau}_i)$ rows of $A$, each set to $a_i$ with probability $p_i$, then with probability at least $1 - \delta$ we have*

$$\min_{\text{rank-}k \ X : \text{row}(X) \subseteq \text{row}(C)} \|A - X\|_F^2 \leq (1 + \epsilon) \|A - A_k\|_F^2 .$$

Recall that the sketch monotonicity for low-rank approximation says that concatenating two sketching matrices $S_1$ and $S_2$ will not increase the error compared to the single sketch matrix $S_1$ or $S_2$, Now matter how $S_1$ and $S_2$ are constructed. (see Section D.1 and Section 4 in (Indyk et al., 2019))

*Proof.* We first consider the first condition. From the condition that $\tau_i(B) \geq \frac{1}{\beta}\tau_i(A)$ we know that if we sample $m = O(\beta \cdot (k \log k + k/\epsilon))$ rows according to $\tau_i(A)$. The actual probability that the $i$-th row of $B$ gets sampled is

$$1 - (1 - \tau_i(A))^m = O(m \cdot \tau_i(A)) = O\left((k \log k + k/\epsilon) \cdot \tau_i(B)\right) .$$

From $\sum_i \tau_i(B) \leq 2k$ and Lemma E.2 (recall the sketch monotonicity property for LRA), we have that with probability at least $9/10$, $S_2$ is a matrix such that

$$\min_{\text{rank-}k \ X : \text{row}(X) \subseteq \text{row}(S_2 B)} \|B - X\|_F^2 \leq (1 + \epsilon) \|B - B_k\|_F^2 .$$

Hence, since $S = \left[\begin{smallmatrix} S_1 \\ S_2 \end{smallmatrix}\right]$, from the the sketch monotonicity property for LRA we have that

$$\min_{\text{rank-}k \ X : \text{row}(X) \subseteq \text{row}(SB)} \|B - X\|_F^2 \leq (1 + \epsilon) \|B - B_k\|_F^2 .$$

Now we consider the second condition. Suppose that $\{X_i\}_{i \leq m}$ and $\{Y_i\}_{i \leq m}$ are a sequence of $m = O(k \log k + k/\epsilon)$ samples from $[n]$ according to the sampling probability distribution $p$ and $q$, where $p_i = \frac{\tau_i(A)}{\sum_i \tau_i(A)}$ and $q_i = \frac{\tau_i(B)}{\sum_i \tau_i(B)}$. Let $S$ be the set of index $i$ such that $X_i \neq Y_i$. From the property of the total variation distance, we get that

$$\Pr[X_i \neq Y_i] \leq d_{\text{tv}}(p, q) = \beta ,$$

and

$$\mathbb{E}[|S|] = \sum_i \Pr[X_i \neq Y_i] \leq \beta m.$$

From Markov's inequality we get that with probability at least $1 - 1.1\beta$, $|S| \leq 1/(1.1\beta) \cdot \beta m = \frac{10}{11}m$. Let $T$ be the set of index $i$ such that $X_i = Y_i$. We have that with probability at least $1 - 1.1\beta$, $|T| \geq m - \frac{10}{11}m = \Omega(k \log k + k/\epsilon)$. Because that $\{Y_i\}_{i \in T}$ is i.i.d samples according to $q$ and the

actual sample we take is $\{X_i\}_{i \in T}$. From Lemma E.2 we get that with probability at least $9/10$, the row space of $B_T$ satisfies

$$\min_{\text{rank-}k \; X:\text{row}(X)\subseteq\text{row}(B_T)} \|B - X\|_F^2 \leq (1 + \epsilon) \|B - B_k\|_F^2 \; .$$

Similarly, from the the sketch monotonicity property we have that with probability at least $0.9 - 1.1\beta$

$$\min_{\text{rank-}k \; X:\text{row}(X)\subseteq\text{row}(SB)} \|B - X\|_F^2 \leq (1 + \epsilon) \|B - B_k\|_F^2 \; . \qquad \square$$

## E.2   PROOF OF THEOREM 6.1

First we prove the following lemma.

**Lemma E.3.** *Let $\delta \in (0, 1/m]$. It holds with probability at least $1 - \delta$ that*

$$\sup_{x \in \text{colsp}(A)} \left| \|Sx\|_2^2 - \|x\|_2^2 \right| \leq \epsilon \|x\|_2^2,$$

*provided that*

$$m \gtrsim \epsilon^{-2}((d + \log m) \min\{\log^2(d/\epsilon), \log^2 m\} + d \log(1/\delta)),$$
$$1 \gtrsim \epsilon^{-2}\nu((\log m) \min\{\log^2(d/\epsilon), \log^2 m\} + \log(1/\delta)) \log(1/\delta).$$

*Proof.* We shall adapt the proof of Theorem 5 in (Bourgain et al., 2015) to our setting. Let $T$ denote the unit sphere in $\text{colsp}(A)$ and set the sparsity parameter $s = 1$. Observe that $\|Sx\|_2^2 = \|x_I\|_2^2 + \|Sx_{I^c}\|_2^2$, and so it suffices to show that

$$\Pr \left\{ \left| \|S'x_{I^c}\|_2^2 - \|x_{I^c}\|_2^2 \right| > \epsilon \right\} \leq \delta$$

for $x \in T$. We make the following definition, as in (2.6) of (Bourgain et al., 2015):

$$A_{\delta,x} := \sum_{i=1}^{m} \sum_{j \in I^c} \delta_{ij} x_j e_i \otimes e_j,$$

and thus, $S'x_{I^c} = A_{\delta,x}\sigma$. Also by $\mathbf{E} \|S'x_{I^c}\|_2^2 = \|x_{I^c}\|_2^2$, one has

$$\sup_{x \in T} \left| \|S'x_{I^c}\|_2^2 - \|x_{I^c}\|_2^2 \right| = \sup_{x \in T} \left| \|A_{\delta,x}\sigma\|_2^2 - \mathbf{E} \|A_{\delta,x}\sigma\|_2^2 \right|. \qquad (\text{E.1})$$

Now, in (2.7) of (Bourgain et al., 2015) we instead define a semi-norm

$$\|x\|_\delta = \max_{1 \leq i \leq m} \left( \sum_{j \in I^c} \delta_{ij} x_j^2 \right)^{1/2}.$$

Then (2.8) continues to hold, and (2.9) as well as (2.10) continue to hold if the supremum in the left-hand side is replaced with the left-hand side of (E.1). At the beginning of Theorem 5, we define $U^{(i)}$ to be $U$, but each row $j \in I^c$ is multiplied by $\delta_{ij}$ and each row $j \in I$ is zeroed out. Then we have in the first step of (4.5) that

$$\sum_{j \in I^c} \delta_{ij} \left| \sum_{k=1}^{d} g_k \langle f_k, e_j \rangle \right|^2 \leq \left\| U^{(i)}g \right\|_2^2,$$

instead of equality. One can verify that the rest of (4.5) goes through. It remains true that $\|\cdot\|_\delta \leq (1/\sqrt{s}) \|\cdot\|_2$, and thus (4.6) holds. One can verify that the rest of the proof of Theorem 5 in (Bourgain et al., 2015) continues to hold if we replace $\sum_{j=1}^{n}$ with $\sum_{j \in I^c}$ and $\max_{1 \leq j \leq n}$ with $\max_{j \in I^c}$, noting that

$$\mathbf{E} \sum_{j \in I^c} \delta_{ij} \|P_E e_j\|_2^2 = \frac{s}{m} \sum_{j \in I^c} \langle P_E e_j, e_j \rangle \leq \frac{s}{m} d$$

and

$$\mathbf{E}(U^{(i)})^* U^{(i)} = \sum_{j \in I^c} (\mathbf{E} \, \delta_{ij}) u_j u_j^* \preceq \frac{1}{m}.$$

Thus, the symmetrization inequalities on

$$\left\| \sum_{j \in I^c} \delta_{ij} \left\| P_E e_j \right\|_2^2 \right\|_{L_\delta^p} \quad \text{and} \quad \left\| \sum_{j \in I^c} \delta_{ij} u_j u_j^* \right\|_{L_\delta^p}$$

continue to hold. The result then follows, observing that $\max_{j \in I^c} \left\| P_E e_j \right\|^2 \leq \nu$. □

The subspace embedding guarantee now follows as a corollary.

**Theorem 6.1.** *Let $\nu = \epsilon/d$. Suppose that $m = \Omega((d/\epsilon^2)(\mathrm{polylog}(1/\epsilon) + \log(1/\delta)))$, $\delta \in (0, 1/m)$ and $d = \Omega((1/\epsilon) \, \mathrm{polylog}(1/\epsilon) \log^2(1/\delta))$. Then, there exists a distribution on $S$ with $m + |I|$ rows such that*

$$\Pr \left\{ \forall x \in \mathrm{colsp}(A), \left| \|Sx\|_2^2 - \|x\|_2^2 \right| > \epsilon \|x\|_2^2 \right\} \leq \delta.$$

*Proof.* One can verify that the two conditions in Lemma E.3 are satisfied if

$$m \gtrsim \frac{d}{\epsilon^2} \left( \mathrm{polylog}(\frac{d}{\epsilon}) + \log \frac{1}{\delta} \right),$$

$$d \gtrsim \frac{1}{\epsilon} \left( \log \frac{1}{\delta} \right) \left( \mathrm{polylog}(\frac{d}{\epsilon}) + \log \frac{1}{\delta} \right).$$

The last condition is satisfied if

$$d \gtrsim \frac{1}{\epsilon} \left( \log^2 \frac{1}{\delta} \right) \mathrm{polylog} \left( \frac{1}{\epsilon} \right). \qquad \square$$

### E.3 PROOF OF LEMMA 6.2

*Proof.* On the one hand, since $Q = SAR$ is an orthogonal matrix, we have

$$\|x\|_2 = \|Qx\|_2 = \|SARx\|_2. \tag{E.2}$$

On the other hand, the assumption implies that

$$\left\| (ARx)^T (ARx) - x^T x \right\|_2 \leq \epsilon \|x\|_2^2,$$

that is,

$$(1 - \epsilon) \|x\|_2^2 \leq \|ARx\|_2^2 \leq (1 + \epsilon) \|x\|_2^2. \tag{E.3}$$

Combining both (E.2) and (E.3) leads to

$$\sqrt{1 - \epsilon} \, \|SARx\|_2 \leq \|ARx\|_2 \leq \sqrt{1 + \epsilon} \, \|SARx\|_2,$$

$$\forall x \in \mathbb{R}^d.$$

Equivalently, it can be written as

$$\frac{1}{\sqrt{1 + \epsilon}} \|SAy\|_2 \leq \|Ay\|_2 \leq \frac{1}{\sqrt{1 - \epsilon}} \|SAy\|_2, \quad \forall y \in \mathbb{R}^d.$$

The claimed result follows from the fact that $1/\sqrt{1 + \epsilon} \geq 1 - \epsilon$ and $1/\sqrt{1 - \epsilon} \leq 1 + \epsilon$ whenever $\epsilon \in (0, \frac{\sqrt{5} - 1}{2}]$. □

# F    LOCATION OPTIMIZATION IN COUNTSKETCH: GREEDY SEARCH

While the position optimization idea is simple, one particularly interesting aspect is that it is provably better than a random placement in some scenarios (Theorem. F.1). Specifically, it is provably beneficial for LRA when inputs follow the spiked covariance model or Zipfian distributions, which are common for real data.

**Spiked covariance model.** Every matrix $A \in \mathbb{R}^{n \times d}$ from the distribution $\mathcal{A}_{sp}(s, \ell)$ has $s < k$ "heavy" rows $A_{r_1}, \cdots, A_{r_s}$ of norm $\ell > 1$. The indices of the heavy rows can be arbitrary, but must be the same for all members of $\mathcal{A}_{sp}(s, \ell)$ and are unknown to the algorithm. The remaining ("light") rows have unit norm. In other words, let $\mathcal{R} = \{r_1, \ldots, r_s\}$. For all rows $A_i, i \in [n]$, $A_i = \ell \cdot v_i$ if $i \in \mathcal{R}$ and $A_i = v_i$ otherwise, where $v_i$ is a uniformly random unit vector.

**Zipfian on squared row norms.** Every $A \in \mathbb{R}^{n \times d} \sim \mathcal{A}_{zipf}$ has rows which are uniformly random and orthogonal. Each $A$ has $2^{i+1}$ rows of squared norm $n^2/2^{2i}$ for $i \in [1, \ldots, \mathcal{O}(\log(n))]$. We also assume that each row has the same squared norm for all members of $\mathcal{A}_{zipf}$.

**Theorem F.1.** *Consider a matrix $A$ from either the spiked covariance model or a Zipfian distribution. Let $S_L$ denote a CountSketch constructed by Algorithm 3 that optimizes the positions of the non-zero values with respect to $A$. Let $S_C$ denote a CountSketch matrix. Then there is a fixed $\eta > 0$ such that,*
$$\min\nolimits_{\text{rank-}k \ X \in \text{rowsp}(S_L A)} \|X - A\|_F^2 \leq (1 - \eta) \min\nolimits_{\text{rank-}k \ X \in \text{rowsp}(S_C A)} \|X - A\|_F^2$$

*Remark* F.2. Note that the above theorem implicitly provides an upper bound on the generalization error of the greedy placement method on the two distributions that we considered in this paper. More precisely, for each of these two distributions, if $\Pi$ is learned via our greedy approach over a set of sampled training matrices, the solution returned by the sketching algorithm using $\Pi$ over any (test) matrix $A$ sampled from the distribution has error at most $(1 - \eta) \min\nolimits_{\text{rank-}k \ X \in \text{rowsp}(S_C A)} \|X - A\|_F^2$.

A key structural property of the matrices from these two distributions that is crucial in our analysis is the $\epsilon$-*almost orthogonality* of their rows (i.e., (normalized) pairwise inner products are at most $\epsilon$). Hence, we can find a QR-factorization of the matrix of such vectors where the upper diagonal matrix $R$ has diagonal entries close to 1 and entries above the diagonal are close to 0.

To state our result, we first provide an interpretation of the location optimization task as a selection of hash function for the rows of $A$. Note that left-multiplying $A$ by CountSketch $S \in \mathbb{R}^{m \times n}$ is equivalent to hashing the rows of $A$ to $m$ bins with coefficients in $\{\pm 1\}$. The greedy algorithm proceeds through the rows of $A$ (in some order) and decides which bin to hash to, denoting this by adding an entry to $S$. The intuition is that our greedy approach separates heavy-norm rows (which are important "directions" in the row space) into different bins.

*Proof Sketch of Theorem F.1* The first step is to observe that in the greedy algorithm, when rows are examined according to a non-decreasing order of squared norms, the algorithm will isolate rows into their singleton bins until all bins are filled. In particular, this means that the heavy norm rows will all be isolated—e.g., for the spiked covariance model, Lemma F.8 presents the formal statement.

Next, we show that none of the rows left to be processed (all light rows) will be assigned to the same bin as a heavy row. The main proof idea is to compare the cost of "colliding" with a heavy row to the cost of "avoiding" the heavy rows. This is the main place we use the properties of the aforementioned distributions and the fact that each heavy row is already mapped to a singleton bin. Overall, we show that at the end of the algorithm no light row will be assigned to the bins that contain heavy rows—the formal statement and proof for the spiked covariance model is in Lemma F.12.

Finally, we can interpret the randomized construction of CountSketch as a "balls and bins" experiment. In particular, considering the heavy rows, we compute the expected number of bins (i.e., rows in $S_C A$) that contain a heavy row. Note that the expected number of rows in $S_C A$ that do not contain any heavy row is $k \cdot (1 - \frac{1}{k})^s \geq k \cdot e^{-\frac{s}{k-1}}$. Hence, the number of rows in $S_C A$ that contain a heavy row of $A$ is at most $k(1 - e^{-\frac{s}{k-1}})$. Thus, at least $s - k(1 - e^{-\frac{s}{k-1}})$ heavy rows are not mapped to an isolated bin (i.e., they collide with some other heavy rows). Then, it is straightforward to show that the squared loss of the solution corresponding to $S_C$ is larger than the squared loss of the solution corresponding to $S_L$, the CountSketch constructed by Algorithm 3—please see Lemma F.14 for the formal statement of its proof.

**Preliminaries and notation.** Left-multiplying $A$ by a CountSketch $S \in \mathbb{R}^{m \times n}$ is equivalent to hashing the rows of $A$ to $m$ bins with coefficients in $\{\pm 1\}$. The greedy algorithm proceeds through the rows of $A$ (in some order) and decides which bin to hash to, which we can think of as adding an entry to $S$. We will denote the bins by $b_i$ and their summed contents by $w_i$.

## F.1 SPIKED COVARIANCE MODEL WITH SPARSE LEFT SINGULAR VECTORS.

To recap, every matrix $A \in \mathbb{R}^{n \times d}$ from the distribution $\mathcal{A}_{sp}(s, \ell)$ has $s < k$ "heavy" rows $(A_{r_1}, \cdots, A_{r_s})$ of norm $\ell > 1$. The indices of the heavy rows can be arbitrary, but must be the same for all members of the distribution and are unknown to the algorithm. The remaining rows (called "light" rows) have unit norm.

In other words: let $\mathcal{R} = \{r_1, \ldots, r_s\}$. For all rows $A_i, i \in [n]$:

$$A_i = \begin{cases} \ell \cdot v_i & \text{if } i \in \mathcal{R} \\ v_i & \text{o.w.} \end{cases}$$

where $v_i$ is a uniformly random unit vector.

We also assume that $S_r, S_g \in \mathbb{R}^{k \times n}$ and that the greedy algorithm proceeds in a non-increasing row norm order.

**Proof sketch.** First, we show that the greedy algorithm using a non-increasing row norm ordering will isolate heavy rows (i.e., each is alone in a bin). Then, we conclude by showing that this yields a better $k$-rank approximation error when $d$ is sufficiently large compared to $n$.

We begin with some preliminary observations that will be of use later.

It is well-known that a set of uniformly random vectors is $\epsilon$-*almost orthogonal* (i.e., the magnitudes of their pairwise inner products are at most $\epsilon$).

*Observation* F.3. Let $v_1, \cdots, v_n \in \mathbb{R}^d$ be a set of random unit vectors. Then with probability $1 - 1/\operatorname{poly}(n)$, we have $|\langle v_i, v_j \rangle| \leq 2\sqrt{\frac{\log n}{d}}, \forall\, i < j \leq n$.

We define $\bar{\epsilon} = 2\sqrt{\frac{\log n}{d}}$.

*Observation* F.4. Let $u_1, \cdots, u_t$ be a set of vectors such that for each pair of $i < j \leq t$, $|\langle \frac{u_i}{\|u_i\|}, \frac{u_j}{\|u_j\|} \rangle| \leq \epsilon$, and $g_i, \cdots, g_j \in \{-1, 1\}$. Then,

$$\sum_{i=1}^{t} \|u_i\|_2^2 - 2\epsilon \sum_{i < j \leq t} \|u_i\|_2 \|u_j\|_2 \leq \left\| \sum_{i=1}^{t} g_i u_i \right\|_2^2 \leq \sum_{i=1}^{t} \|u_i\|_2^2 + 2\epsilon \sum_{i < j \leq t} \|u_i\|_2 \|u_j\|_2 \quad \text{(F.1)}$$

Next, a straightforward consequence of $\epsilon$-almost orthogonality is that we can find a QR-factorization of the matrix of such vectors where $R$ (an upper diagonal matrix) has diagonal entries close to $1$ and entries above the diagonal are close to $0$.

**Lemma F.5.** *Let $u_1, \cdots, u_t \in \mathbb{R}^d$ be a set of unit vectors such that for any pair of $i < j \leq t$, $|\langle u_i, u_j \rangle| \leq \epsilon$ where $\epsilon = O(t^{-2})$. There exists an orthonormal basis $e_1, \cdots, e_t$ for the subspace spanned by $u_1, \cdots, u_t$ such that for each $i \leq t$, $u_i = \sum_{j=1}^{i} a_{i,j} e_j$ where $a_{i,i}^2 \geq 1 - \sum_{j=1}^{i-1} j^2 \cdot \epsilon^2$ and for each $j < i$, $a_{i,j}^2 \leq j^2 \epsilon^2$.*

*Proof.* We follow the Gram-Schmidt process to construct the orthonormal basis $e_1, \cdots, e_t$ of the space spanned by $u_1, \cdots, u_t$, by first setting $e_1 = u_1$ and then processing $u_2, \cdots, u_t$, one-by-one.

The proof is by induction. We show that once the first $j$ vectors $u_1, \cdots, u_j$ are processed, the statement of the lemma holds for these vectors. Note that the base case of the induction trivially holds as $u_1 = e_1$. Next, suppose that the induction hypothesis holds for the first $\ell$ vectors $u_1, \cdots, u_\ell$.

*Claim* F.6. For each $j \leq \ell$, $a_{\ell+1,j}^2 \leq j^2 \epsilon^2$.

*Proof.* The proof of the claim is itself by induction. Note that, for $j = 1$ and using the fact that $|\langle u_1, u_{\ell+1} \rangle| \leq \epsilon$, the statement holds and $a_{\ell+1,1}^2 \leq \epsilon^2$. Next, suppose that the statement holds for all

$j \leq i < \ell$. Then using that $|\langle u_{i+1}, u_{\ell+1}\rangle| \leq \epsilon$,

$$|a_{\ell+1,i+1}| \leq (|\langle u_{\ell+1}, u_{i+1}\rangle| + \sum_{j=1}^{i} |a_{\ell+1,j}| \cdot |a_{i+1,j}|)/|a_{i+1,i+1}|$$

$$\leq (\epsilon + \sum_{j=1}^{i} j^2 \epsilon^2)/|a_{i+1,i+1}| \quad \triangleright \text{ by the induction hypothesis on } a_{\ell+1,j} \text{ for } j \leq i$$

$$\leq (\epsilon + \sum_{j=1}^{i} j^2 \epsilon^2)/(1 - \sum_{j=1}^{i} j^2 \cdot \epsilon^2)^{1/2} \quad \triangleright \text{ by the induction hypothesis on } a_{i+1,i+1}$$

$$\leq (\epsilon + \sum_{j=1}^{i} j^2 \epsilon^2) \cdot (1 - \sum_{j=1}^{i} j^2 \cdot \epsilon^2)^{1/2} \cdot (1 + 2 \cdot \sum_{j=1}^{i} j^2 \epsilon^2)$$

$$\leq (\epsilon + \sum_{j=1}^{i} j^2 \epsilon^2) \cdot (1 + 2 \cdot \sum_{j=1}^{i} j^2 \epsilon^2)$$

$$\leq \epsilon((\sum_{j=1}^{i} j^2 \epsilon) \cdot (1 + 4\epsilon \cdot \sum_{j=1}^{i} j^2 \epsilon) + 1)$$

$$\leq \epsilon(i + 1) \quad \triangleright \text{ by } \epsilon = O(t^{-2})$$

$\square$

Finally, since $\|u_{\ell+1}\|_2^2 = 1$, $a_{\ell+1,\ell+1}^2 \geq 1 - \sum_{j=1}^{\ell} j^2 \epsilon^2$. $\square$

**Corollary F.7.** *Suppose that $\bar{\epsilon} = O(t^{-2})$. There exists an orthonormal basis $e_1, \cdots, e_t$ for the space spanned by the randomly picked vectors $v_1, \cdots, v_t$, of unit norm, so that for each $i$, $v_i = \sum_{j=1}^{i} a_{i,j} e_j$ where $a_{i,i}^2 \geq 1 - \sum_{j=1}^{i-1} j^2 \cdot \bar{\epsilon}^2$ and for each $j < i$, $a_{i,j}^2 \leq j^2 \cdot \bar{\epsilon}^2$.*

*Proof.* The proof follows from Lemma F.5 and the fact that the set of vectors $v_1, \cdots, v_t$ is $\bar{\epsilon}$-almost orthogonal (by Observation F.3). $\square$

The first main step is to show that the greedy algorithm (with non-increasing row norm ordering) will isolate rows into their own bins until all bins are filled. In particular, this means that the heavy rows (the first to be processed) will all be isolated.

We note that because we set $\text{rank}(SA) = k$, the $k$-rank approximation cost is the simplified expression $\left\|AVV^\top - A\right\|_F^2$, where $U\Sigma V^\top = SA$, rather than $\left\|[AV]_k V^\top - A\right\|_F^2$. This is just the projection cost onto $\text{row}(SA)$. Also, we observe that minimizing this projection cost is the same as maximizing the sum of squared projection coefficients:

$$\arg\min_S \left\|A - AVV^\top\right\|_F^2 = \arg\min_S \sum_{i \in [n]} \|A_i - (\langle A_i, v_1\rangle v_1 + \ldots + \langle A_i, v_k\rangle v_k)\|_2^2$$

$$= \arg\min_S \sum_{i \in [n]} (\|A_i\|_2^2 - \sum_{j \in [k]} \langle A_i, v_j\rangle^2)$$

$$= \arg\max_S \sum_{i \in [n]} \sum_{j \in [k]} \langle A_i, v_j\rangle^2$$

In the following sections, we will prove that our greedy algorithm makes certain choices by showing that these choices maximize the sum of squared projection coefficients.

**Lemma F.8.** *For any matrix $A$ or batch of matrices $\mathcal{A}$, at the end of iteration $k$, the learned CountSketch matrix $S$ maps each row to an isolated bin. In particular, heavy rows are mapped to isolated bins.*

*Proof.* For any iteration $i \leq k$, we consider the choice of assigning $A_i$ to an empty bin versus an occupied bin. Without loss of generality, let this occupied bin be $b_{i-1}$, which already contains $A_{i-1}$.

We consider the difference in cost for empty versus occupied. We will do this cost comparison for $A_j$ with $j \leq i-2$, $j \geq i+1$, and finally, $j \in \{i-1, i\}$.

First, we let $\{e_1, \ldots, e_i\}$ be an orthonormal basis for $\{A_1, \ldots, A_i\}$ such that for each $r \leq i$, $A_r = \sum_{j=1}^{r} a_{r,j} e_j$ where $a_{r,r} > 0$. This exists by Lemma F.5. Let $\{e_1, \ldots, e_{i-2}, \overline{e}\}$ be an orthonormal basis for $\{A_1, \ldots, A_{i+2}, A_{i-1} \pm A_i\}$. Now, $\overline{e} = c_0 e_{i-1} + c_1 e_i$ for some $c_0, c_1$ because $(A_{i-1} \pm A_i) - \text{proj}_{\{e_1, \ldots, e_{i-2}\}}(A_{i-1} \pm A_i) \in \text{span}(e_{i-1}, e_i)$. We note that $c_0^2 + c_1^2 = 1$ because we let $\overline{e}$ be a unit vector. We can find $c_0, c_1$ to be:

$$c_0 = \frac{a_{i-1,i-1} + a_{i,i-1}}{\sqrt{(a_{i-1,i-1} + a_{i,i-1})^2 + a_{i,i}^2}}, \quad c_1 = \frac{a_{i,i}}{\sqrt{(a_{i-1,i-1} + a_{i,i-1})^2 + a_{i,i}^2}}$$

1. $j \leq i-2$: The cost is zero for both cases because $A_j \in \text{span}(\{e_1, \ldots, e_{i-2}\})$.

2. $j \geq i+1$: We compare the rewards (sum of squared projection coefficients) and find that $\{e_1, \ldots, e_{i-2}, \overline{e}\}$ is no better than $\{e_1, \ldots, e_i\}$.

$$\begin{aligned} \langle A_j, \overline{e} \rangle^2 &= (c_0 \langle A_j, e_{i-1} \rangle + c_1 \langle A_j, e_i \rangle)^2 \\ &\leq (c_1^2 + c_0^2)(\langle A_j, e_{i-1} \rangle^2 + \langle A_j, e_i \rangle^2) \qquad \triangleright \text{ Cauchy-Schwarz inequality} \\ &= \langle A_j, e_{i-1} \rangle^2 + \langle A_j, e_i \rangle^2 \end{aligned}$$

3. $j \in \{i-1, i\}$: We compute the sum of squared projection coefficients of $A_{i-1}$ and $A_i$ onto $\overline{e}$:

$$\begin{aligned} &(\frac{1}{(a_{i-1,i-1} + a_{i,i-1})^2 + a_{i,i}^2}) \cdot (a_{i-1,i-1}^2(a_{i-1,i-1} + a_{i,i-1})^2 + (a_{i,i-1}(a_{i-1,i-1} + a_{i,i-1}) + a_{i,i}a_{i,i})^2) \\ &= (\frac{1}{(a_{i-1,i-1} + a_{i,i-1})^2 + a_{i,i}^2}) \cdot ((a_{i-1,i-1} + a_{i,i-1})^2(a_{i-1,i-1}^2 + a_{i,i-1}^2) \\ &\quad + a_{i,i}^4 + 2a_{i,i-1}a_{i,i}^2(a_{i-1,i-1} + a_{i,i-1})) \end{aligned} \qquad \text{(F.2)}$$

On the other hand, the sum of squared projection coefficients of $A_{i-1}$ and $A_i$ onto $e_{i-1} \cup e_i$ is:

$$(\frac{(a_{i-1,i-1} + a_{i,i-1})^2 + a_{i,i}^2}{(a_{i-1,i-1} + a_{i,i-1})^2 + a_{i,i}^2}) \cdot (a_{i-1,i-1}^2 + a_{i,i-1}^2 + a_{i,i}^2) \qquad \text{(F.3)}$$

Hence, the difference between the sum of squared projections of $A_{i-1}$ and $A_i$ onto $\overline{e}$ and $e_{i-1} \cup e_i$ is ((F.3) - (F.2))

$$\begin{aligned} &\frac{a_{i,i}^2((a_{i-1,i-1} + a_{i,i-1})^2 + a_{i-1,i-1}^2 + a_{i,i-1}^2 - 2a_{i,i-1}(a_{i-1,i-1} + a_{i,i-1}))}{(a_{i-1,i-1} + a_{i,i-1})^2 + a_{i,i}^2} \\ &= \frac{2a_{i,i}^2 a_{i-1,i-1}^2}{(a_{i-1,i-1} + a_{i,i-1})^2 + a_{i,i}^2} > 0 \end{aligned}$$

Thus, we find that $\{e_1, \ldots, e_i\}$ is a strictly better basis than $\{e_1, \ldots, e_{i-2}, \overline{e}\}$. This means the greedy algorithm will choose to place $A_i$ in an empty bin. $\qquad \square$

Next, we show that none of the rows left to be processed (all light rows) will be assigned to the same bin as a heavy row. The main proof idea is to compare the cost of "colliding" with a heavy row to the cost of "avoiding" the heavy rows. Specifically, we compare the *decrease* (before and after bin assignment of a light row) in sum of squared projection coefficients, lower-bounding it in the former case and upper-bounding it in the latter.

We introduce some results that will be used in Lemma F.12.

*Claim* F.9. Let $A_{k+r}, r \in [1, \ldots, n-k]$ be a light row not yet processed by the greedy algorithm. Let $\{e_1, \ldots, e_k\}$ be the Gram-Schmidt basis for the current $\{w_1, \ldots, w_k\}$. Let $\beta = \mathcal{O}(n^{-1}k^{-3})$ upper bound the inner products of the normalized $A_{k+r}, w_1, \ldots, w_k$. Then, for any bin $i$, $\langle e_i, A_{k+r} \rangle^2 \leq \beta^2 \cdot k^2$.

*Proof.* This is a straightforward application of Lemma F.5. From that, we have $\langle A_{k+r}, e_i\rangle^2 \le i^2\beta^2$, for $i \in [1, \ldots, k]$, which means $\langle A_{k+r}, e_i\rangle^2 \le k^2\beta^2$. $\qquad\square$

*Claim* F.10. Let $A_{k+r}$ be a light row that has been processed by the greedy algorithm. Let $\{e_1, \ldots, e_k\}$ be the Gram-Schmidt basis for the current $\{w_1, \ldots, w_k\}$. If $A_{k+r}$ is assigned to bin $b_{k-1}$ (w.l.o.g.), the squared projection coefficient of $A_{k+r}$ onto $e_i, i \ne k-1$ is at most $4\beta^2 \cdot k^2$, where $\beta = \mathcal{O}(n^{-1}k^{-3})$ upper bounds the inner products of normalized $A_{k+r}, w_1, \cdots, w_k$.

*Proof.* Without loss of generality, it suffices to bound the squared projection of $A_{k+r}$ onto the direction of $w_k$ that is orthogonal to the subspace spanned by $w_1, \cdots, w_{k-1}$. Let $e_1, \cdots, e_k$ be an orthonormal basis of $w_1, \cdots, w_k$ guaranteed by Lemma F.5. Next, we expand the orthonormal basis to include $e_{k+1}$ so that we can write the normalized vector of $A_{k+r}$ as $v_{k+r} = \sum_{j=1}^{k+1} b_j e_j$. By a similar approach to the proof of Lemma F.5, for each $j \le k - 2$, $b_j \le \beta^2 j^2$. Next, since $|\langle w_k, v_{k+r}\rangle| \le \beta$,

$$
\begin{aligned}
|b_k| &\le \frac{1}{|\langle w_k, e_k\rangle|} \cdot \left(|\langle w_k, v_{k+r}\rangle| + \sum_{j=1}^{k-1} |b_j \cdot \langle w_k, e_j\rangle|\right) \\
&\le \frac{1}{\sqrt{1 - \sum_{j=1}^{k-1} \beta^2 \cdot j^2}} \cdot \left(\beta + \sum_{j=1}^{k-2} \beta^2 \cdot j^2 + (k-1) \cdot \beta\right) \quad \triangleright\ |b_{k-1}| \le 1 \\
&= \frac{\beta + \sum_{j=1}^{k-2} \beta^2 \cdot j^2}{\sqrt{1 - \sum_{j=1}^{k-1} \beta^2 \cdot j^2}} + (k-1)\beta \\
&\le 2(k-1)\beta - \frac{\beta^2 (k-1)^2}{\sqrt{1 - \sum_{j=1}^{k-1} \beta^2 \cdot j^2}} \quad\quad\quad\quad \triangleright\ \text{similar to the proof of Lemma F.5} \\
&< 2\beta \cdot k
\end{aligned}
$$

Hence, the squared projection of $A_{k+r}$ onto $e_k$ is at most $4\beta^2 \cdot k^2 \cdot \|A_{k+r}\|_2^2$. We assumed $\|A_{k+r}\| = 1$; hence, the squared projection of $A_{k+r}$ onto $e_k$ is at most $4\beta^2 \cdot k^2$. $\qquad\square$

*Claim* F.11. We assume that the absolute values of the inner products of vectors in $v_1, \cdots, v_n$ are at most $\bar{\epsilon} < 1/(n^2 \sum_{A_i \in b} \|A_i\|_2)$ and the absolute values of the inner products of the normalized vectors of $w_1, \cdots, w_k$ are at most $\beta = \mathcal{O}(n^{-3}k^{-\frac{3}{2}})$. Suppose that bin $b$ contains the row $A_{k+r}$. Then, the squared projection of $A_{k+r}$ onto the direction of $w$ orthogonal to $\mathsf{span}(\{w_1, \cdots, w_k\} \setminus \{w\})$ is at most $\frac{\|A_{k+r}\|_2^4}{\|w\|_2^2} + \mathcal{O}(n^{-2})$ and is at least $\frac{\|A_{k+r}\|_2^4}{\|w\|_2^2} - \mathcal{O}(n^{-2})$.

*Proof.* Without loss of generality, we assume that $A_{k+r}$ is mapped to $b_k$; $w = w_k$. First, we provide an upper and a lower bound for $|\langle v_{k+r}, \overline{w}_k\rangle|$ where for each $i \le k$, we let $\overline{w}_i = \frac{w_i}{\|w_i\|_2}$ denote the normalized vector of $w_i$. Recall that by definition $v_{k+r} = \frac{A_{k+r}}{\|A_{k+r}\|_2}$.

$$
\begin{aligned}
|\langle \overline{w}_k, v_{k+r}\rangle| &\le \frac{\|A_{k+r}\|_2 + \sum_{A_i \in b_k} \bar{\epsilon} \|A_i\|_2}{\|w_k\|_2} \\
&\le \frac{\|A_{k+r}\|_2 + n^{-2}}{\|w_k\|_2} \quad\quad\quad \triangleright\ \text{by}\ \bar{\epsilon} < \frac{n^{-2}}{\sum_{A_i \in b_k} \|A_i\|_2} \\
&\le \frac{\|A_{k+r}\|_2}{\|w_k\|_2} + n^{-2} \quad\quad\quad \triangleright\ \|w_k\|_2 \ge 1 \quad\quad\quad\quad (\text{F.4})
\end{aligned}
$$

$$
\begin{aligned}
|\langle \overline{w}_k, v_{k+r}\rangle| &\ge \frac{\|A_{k+r}\|_2 - \sum_{A_i \in b_k} \|A_i\|_2 \cdot \bar{\epsilon}}{\|w_k\|_2} \\
&\ge \frac{\|A_{k+r}\|_2}{\|w_k\|_2} - n^{-2} \quad\quad\quad\quad\quad\quad\quad\quad\quad\quad\quad\quad (\text{F.5})
\end{aligned}
$$

Now, let $\{e_1, \cdots, e_k\}$ be an orthonormal basis for the subspace spanned by $\{w_1, \cdots, w_k\}$ guaranteed by Lemma F.5. Next, we expand the orthonormal basis to include $e_{k+1}$ so that we can write $v_{k+r} = \sum_{j=1}^{k+1} b_j e_j$. By a similar approach to the proof of Lemma F.5, we can show that for each $j \le k-1$, $b_j^2 \le \beta^2 j^2$. Moreover,

$$|b_k| \le \frac{1}{|\langle \overline{w}_k, e_k \rangle|} \cdot (|\langle \overline{w}_k, v_{k+r} \rangle| + \sum_{j=1}^{k-1} |b_j \cdot \langle \overline{w}_k, e_j \rangle|)$$

$$\le \frac{1}{\sqrt{1 - \sum_{j=1}^{k-1} \beta^2 \cdot j^2}} \cdot (|\langle \overline{w}_k, v_{k+r} \rangle| + \sum_{j=1}^{k-1} \beta^2 \cdot j^2) \qquad \triangleright \text{ by Lemma F.5}$$

$$\le \frac{1}{\sqrt{1 - \sum_{j=1}^{k-1} \beta^2 \cdot j^2}} \cdot (n^{-2} + \frac{\|A_{k+r}\|_2}{\|w_k\|_2} + \sum_{j=1}^{k-1} \beta^2 \cdot j^2) \qquad \triangleright \text{ by (F.4)}$$

$$< \beta \cdot k + \frac{1}{\sqrt{1 - \beta^2 k^3}} \cdot (n^{-2} + \frac{\|A_{k+r}\|_2}{\|w_k\|_2}) \qquad \triangleright \text{ similar to the proof of Lemma F.5}$$

$$\le \mathcal{O}(n^{-2}) + (1 + \mathcal{O}(n^{-2})) \frac{\|A_{k+r}\|_2}{\|w_k\|_2} \qquad \triangleright \text{ by } \beta = \mathcal{O}(n^{-3} k^{-\frac{3}{2}})$$

$$\le \frac{\|A_{k+r}\|_2}{\|w_k\|_2} + \mathcal{O}(n^{-2}) \qquad \triangleright \frac{\|A_{k+r}\|_2}{\|w_k\|_2} \le 1$$

and,

$$|b_k| \ge \frac{1}{|\langle \overline{w}_k, e_k \rangle|} \cdot (|\langle \overline{w}_k, v_{k+r} \rangle| - \sum_{j=1}^{k-1} |b_j \cdot \langle \overline{w}_k, e_j \rangle|)$$

$$\ge |\langle \overline{w}_k, v_{k+r} \rangle| - \sum_{j=1}^{k-1} \beta^2 \cdot j^2 \qquad \triangleright \text{ since } |\langle \overline{w}_k, e_k \rangle| \le 1$$

$$\ge \frac{\|A_{k+r}\|_2}{\|w_k\|_2} - n^{-2} - \sum_{j=1}^{k-1} \beta^2 \cdot j^2 \qquad \triangleright \text{ by (F.5)}$$

$$\ge \frac{\|A_{k+r}\|_2}{\|w_k\|_2} - \mathcal{O}(n^{-2}) \qquad \triangleright \text{ by } \beta = \mathcal{O}(n^{-3} k^{-\frac{3}{2}})$$

Hence, the squared projection of $A_{k+r}$ onto $e_k$ is at most $\frac{\|A_{k+r}\|_2^4}{\|w_k\|_2^2} + \mathcal{O}(n^{-2})$ and is at least $\frac{\|A_{k+r}\|_2^4}{\|w_k\|_2^2} - \mathcal{O}(n^{-2})$. $\qquad \square$

Now, we show that at the end of the algorithm no light row will be assigned to the bins that contain heavy rows.

**Lemma F.12.** *We assume that the absolute values of the inner products of vectors in $v_1, \cdots, v_n$ are at most $\overline{\epsilon} < \min\{n^{-2} k^{-\frac{5}{3}}, (n \sum_{A_i \in w} \|A_i\|_2)^{-1}\}$. At iteration $k+r$, the greedy algorithm will assign the light row $A_{k+r}$ to a bin that does not contain a heavy row.*

*Proof.* The proof is by induction. Lemma F.8 implies that no light row has been mapped to a bin that contains a heavy row for the first $k$ iterations. Next, we assume that this holds for the first $k+r-1$ iterations and show that is also must hold for the $(k+r)$-th iteration.

To this end, we compare the sum of squared projection coefficients when $A_{k+r}$ avoids and collides with a heavy row.

First, we upper bound $\beta = \max_{i \neq j \leq k} |\langle w_i, w_j \rangle| / (\|w_i\|_2 \|w_j\|_2)$. Let $c_i$ and $c_j$ respectively denote the number of rows assigned to $b_i$ and $b_j$.

$$\beta = \max_{i \neq j \leq k} \frac{|\langle w_i, w_j \rangle|}{\|w_i\|_2 \|w_j\|_2} \leq \frac{c_i \cdot c_j \cdot \overline{\epsilon}}{\sqrt{c_i - 2\overline{\epsilon} c_i^2} \cdot \sqrt{c_j - 2\overline{\epsilon} c_j^2}} \qquad \triangleright \text{ Observation F.4}$$

$$\leq 16 \overline{\epsilon} \sqrt{c_i c_j} \qquad\qquad \triangleright \overline{\epsilon} \leq n^{-2} k^{-5/3}$$

$$\leq n^{-1} k^{-\frac{5}{3}} \qquad\qquad \triangleright \overline{\epsilon} \leq n^{-2} k^{-5/3}$$

**1. If $A_{k+r}$ is assigned to a bin that contains $c$ light rows and no heavy rows.** In this case, the projection loss of the heavy rows $A_1, \cdots, A_s$ onto $\text{row}(SA)$ remains zero. Thus, we only need to bound the change in the sum of squared projection coefficients of the light rows before and after iteration $k + r$. Without loss of generality, let $w_k$ denote the bin that contains $A_{k+r}$. Since $\mathcal{S}_{k-1} = \text{span}(\{w_1, \cdots, w_{k-1}\})$ has not changed, we only need to bound the difference in cost between projecting onto the component of $w_k - A_{k+r}$ orthogonal to $\mathcal{S}_{k-1}$ and the component of $w_k$ orthogonal to $\mathcal{S}_{k-1}$, respectively denoted as $e_k$ and $\overline{e}_k$.

1. By Claim F.9, for the light rows that are not yet processed (i.e., $A_j$ for $j > k + r$), the squared projection of each onto $e_k$ is at most $\beta^2 k^2$. Hence, the total decrease in the squared projection is at most $(n - k - r) \cdot \beta^2 k^2$.

2. By Claim F.10, for the processed light rows that are not mapped to the last bin, the squared projection of each onto $e_k$ is at most $4\beta^2 k^2$. Hence, the total decrease in the squared projection cost is at most $(r - 1) \cdot 4\beta^2 k^2$.

3. For each row $A_i \neq A_{k+r}$ that is mapped to the last bin, by Claim F.11 and the fact $\|A_i\|_2^4 = \|A_i\|_2^2 = 1$, the squared projection of $A_i$ onto $e_k$ is at most $\frac{\|A_i\|_2^2}{\|w_k - A_{k+r}\|_2^2} + \mathcal{O}(n^{-2})$ and the squared projection of $A_i$ onto $\overline{e}_k$ is at least $\frac{\|A_i\|_2^2}{\|w_k\|_2^2} - \mathcal{O}(n^{-2})$.

   Moreover, the squared projection of $A_{k+r}$ onto $e_k$ compared to $\overline{e}_k$ increases by at least $\left( \frac{\|A_{k+r}\|_2^2}{\|w_k\|_2^2} - \mathcal{O}(n^{-2}) \right) - \mathcal{O}(n^{-2}) = \frac{\|A_{k+r}\|_2^2}{\|w_k\|_2^2} - \mathcal{O}(n^{-2})$.

   Hence, the total squared projection of the rows in the bin $b_k$ decreases by at least:

$$\left( \sum_{A_i \in w_k / \{A_{r+k}\}} \frac{\|A_i\|_2^2}{\|w_k - A_{r+k}\|_2^2} + \mathcal{O}(n^{-2}) \right) - \left( \sum_{A_i \in w_k} \frac{\|A_i\|_2^2}{\|w_k\|_2^2} - \mathcal{O}(n^{-2}) \right)$$

$$\leq \frac{\|w_k - A_{r+k}\|_2^2 + \mathcal{O}(n^{-1})}{\|w_k - A_{r+k}\|_2^2} - \frac{\|w_k\|_2^2 - \mathcal{O}(n^{-1})}{\|w_k\|_2^2} + \mathcal{O}(n^{-1}) \qquad \triangleright \text{ by Observation F.4}$$

$$\leq \mathcal{O}(n^{-1})$$

Hence, summing up the bounds in items 1 to 3 above, the total decrease in the sum of squared projection coefficients is at most $\mathcal{O}(n^{-1})$.

**2. If $A_{k+r}$ is assigned to a bin that contains a heavy row.** Without loss of generality, we can assume that $A_{k+r}$ is mapped to $b_k$ that contains the heavy row $A_s$. In this case, the distance of heavy rows $A_1, \cdots, A_{s-1}$ onto the space spanned by the rows of $SA$ is zero. Next, we bound the amount of change in the squared distance of $A_s$ and light rows onto the space spanned by the rows of $SA$.

Note that the $(k - 1)$-dimensional space corresponding to $w_1, \cdots, w_{k-1}$ has not changed. Hence, we only need to bound the decrease in the projection distance of $A_{k+r}$ onto $\overline{e}_k$ compared to $e_k$ (where $\overline{e}_k, e_k$ are defined similarly as in the last part).

1. For the light rows other than $A_{k+r}$, the squared projection of each onto $e_k$ is at most $\beta^2 k^2$. Hence, the total increase in the squared projection of light rows onto $e_k$ is at most $(n - k) \cdot \beta^2 k^2 = \mathcal{O}(n^{-1})$.

2. By Claim F.11, the sum of squared projections of $A_s$ and $A_{k+r}$ onto $e_k$ decreases by at least

$$\|A_s\|_2^2 - \left(\frac{\|A_s\|_2^4 + \|A_{k+r}\|_2^4}{\|A_s + A_{r+k}\|_2^2} + \mathcal{O}(n^{-1})\right)$$

$$\geq \|A_s\|_2^2 - \left(\frac{\|A_s\|_2^4 + \|A_{k+r}\|_2^4}{\|A_s\|_2^2 + \|A_{r+k}\|_2^2 - n^{-\mathcal{O}(1)}} + \mathcal{O}(n^{-1})\right) \qquad \triangleright \text{ by Observation F.4}$$

$$\geq \frac{\|A_{r+k}\|_2^2 \left(\|A_s\|_2^2 - \|A_{k+r}\|_2^2\right) - \|A_s\|_2^2 \cdot \mathcal{O}(n^{-1})}{\|A_s\|_2^2 + \|A_{r+k}\|_2^2 - \mathcal{O}(n^{-1})} - \mathcal{O}(n^{-1})$$

$$\geq \frac{\|A_{r+k}\|_2^2 \left(\|A_s\|_2^2 - \|A_{k+r}\|_2^2\right) - \|A_s\|_2^2 \cdot \mathcal{O}(n^{-1})}{\|A_s\|_2^2 + \|A_{r+k}\|_2^2} - \mathcal{O}(n^{-1})$$

$$\geq \frac{\|A_{r+k}\|_2^2 \left(\|A_s\|_2^2 - \|A_{k+r}\|_2^2\right)}{\|A_s\|_2^2 + \|A_{r+k}\|_2^2} - \mathcal{O}(n^{-1})$$

$$\geq \frac{\|A_{r+k}\|_2^2 \left(1 - \left(\|A_{k+r}\|_2^2 / \|A_s\|_2^2\right)\right)}{1 + \left(\|A_{r+k}\|_2^2 / \|A_s\|_2^2\right)} - \mathcal{O}(n^{-1})$$

$$\geq \|A_{r+k}\|_2^2 \left(1 - \frac{\|A_{k+r}\|_2}{\|A_s\|_2}\right) - \mathcal{O}(n^{-1}) \qquad \triangleright \frac{1 - \epsilon^2}{1 + \epsilon^2} \geq 1 - \epsilon$$

Hence, in this case, the total decrease in the squared projection is at least

$$\|A_{r+k}\|_2^2 \left(1 - \frac{\|A_{k+r}\|_2}{\|A_s\|_2}\right) - \mathcal{O}(n^{-1}) = 1 - \frac{\|A_{k+r}\|_2}{\|A_s\|_2}) - \mathcal{O}(n^{-1}) \qquad \triangleright \|A_{r+k}\|_2 = 1$$

$$= 1 - (1/\sqrt{\ell}) - \mathcal{O}(n^{-1}) \qquad \triangleright \|A_s\|_2 = \sqrt{\ell}$$

Thus, for a sufficiently large value of $\ell$, the greedy algorithm will assign $A_{k+r}$ to a bin that only contains light rows. This completes the inductive proof and in particular implies that at the end of the algorithm, heavy rows are assigned to isolated bins. $\square$

**Corollary F.13.** *The approximation loss of the best* rank-$k$ *approximate solution in the rowspace $S_g A$ for $A \sim \mathcal{A}_{sp}(s, \ell)$, where $A \in \mathbb{R}^{n \times d}$ for $d = \Omega(n^4 k^4 \log n)$ and $S_g$ is the CountSketch constructed by the greedy algorithm with non-increasing order, is at most $n - s$.*

*Proof.* First, we need to show that the absolute values of the inner products of vectors in $v_1, \cdots, v_n$ are at most $\bar{\epsilon} < \min\{n^{-2}k^{-2}, (n \sum_{A_i \in w} \|A_i\|_2)^{-1}\}$ so that we can apply Lemma F.12. To show this, note that by Observation F.3, $\bar{\epsilon} \leq 2\sqrt{\frac{\log n}{d}} \leq n^{-2} k^{-2}$ since $d = \Omega(n^4 k^4 \log n)$. The proof follows from Lemma F.8 and Lemma F.12. Since all heavy rows are mapped to isolated bins, the projection loss of the light rows is at most $n - s$. $\square$

Next, we bound the Frobenius norm error of the best rank-$k$-approximation solution constructed by the standard CountSketch with a randomly chosen sparsity pattern.

**Lemma F.14.** *Let $s = \alpha k$ where $0.7 < \alpha < 1$. The expected squared loss of the best* rank-$k$ *approximate solution in the rowspace $S_r A$ for $A \in \mathbb{R}^{n \times d} \sim \mathcal{A}_{sp}(s, \ell)$, where $d = \Omega(n^6 \ell^2)$ and $S_r$ is the sparsity pattern of CountSketch is chosen uniformly at random, is at least $n + \frac{\ell k}{4e} - (1 + \alpha)k - n^{-\mathcal{O}(1)}$.*

*Proof.* We can interpret the randomized construction of the CountSketch as a "balls and bins" experiment. In particular, considering the heavy rows, we compute the expected number of bins (i.e., rows in $S_r A$) that contain a heavy row. Note that the expected number of rows in $S_r A$ that do not contain any heavy row is $k \cdot (1 - \frac{1}{k})^s \geq k \cdot e^{-\frac{s}{k-1}}$. Hence, the number of rows in $S_r A$ that contain a heavy row of $A$ is at most $k(1 - e^{-\frac{s}{k-1}})$. Thus, at least $s - k(1 - e^{-\frac{s}{k-1}})$ heavy rows are not mapped to an isolated bin (i.e., they collide with some other heavy rows). Then, it is straightforward to show that the squared loss of each such row is at least $\ell - n^{-\mathcal{O}(1)}$.

*Claim* F.15. Suppose that heavy rows $A_{r_1}, \cdots, A_{r_c}$ are mapped to the same bin via a CountSketch $S$. Then, the total squared distances of these rows from the subspace spanned by $SA$ is at least $(c-1)\ell - \mathcal{O}(n^{-1})$.

*Proof.* Let $b$ denote the bin that contains the rows $A_{r_1}, \cdots, A_{r_c}$ and suppose that it has $c'$ light rows as well. Note that by Claim F.10 and Claim F.11, the squared projection of each row $A_{r_i}$ onto the subspace spanned by the $k$ bins is at most

$$
\frac{\|A_{h_i}\|_2^4}{\|w\|_2^2} + \mathcal{O}(n^{-1})
$$

$$
\leq \frac{\ell^2}{c\ell + c' - 2\bar{\epsilon}(c^2\ell + cc'\sqrt{\ell} + c'^2)} + \mathcal{O}(n^{-1})
$$

$$
\leq \frac{\ell^2}{c\ell - n^{-\mathcal{O}(1)}} + n^{-\mathcal{O}(1)} \qquad \qquad \rhd \text{ by } \bar{\epsilon} \leq n^{-3}\ell^{-1}
$$

$$
\leq \frac{\ell^2}{c^2\ell^2} \cdot (c\ell + \mathcal{O}(n^{-1}) + \mathcal{O}(n^{-1})
$$

$$
\leq \frac{\ell}{c} + \mathcal{O}(n^{-1})
$$

Hence, the total squared loss of these $c$ heavy rows is at least $c\ell - c \cdot (\frac{\ell}{c} + \mathcal{O}(n^{-1})) \geq (c-1)\ell - \mathcal{O}(n^{-1})$. $\qquad \square$

Thus, the expected total squared loss of the heavy rows is at least:

$$
\ell \cdot \left(s - k(1 - e^{-\frac{s}{k-1}})\right) - s \cdot n^{-\mathcal{O}(1)}
$$

$$
\geq \ell \cdot k(\alpha - 1 + e^{-\alpha}) - \ell\alpha - n^{-\mathcal{O}(1)} \qquad \rhd s = \alpha \cdot (k-1) \text{ where } 0.7 < \alpha < 1
$$

$$
\geq \frac{\ell k}{2e} - \ell - n^{-\mathcal{O}(1)} \qquad \qquad \rhd \alpha \geq 0.7
$$

$$
\geq \frac{\ell k}{4e} - \mathcal{O}(n^{-1}) \qquad \qquad \rhd \text{ assuming } k > 4e
$$

Next, we compute a lower bound on the expected squared loss of the light rows. Note that Claim F.10 and Claim F.11 imply that when a light row collides with other rows, its contribution to the total squared loss (note that the loss accounts for the amount it decreases from the squared projection of the other rows in the bin as well) is at least $1 - \mathcal{O}(n^{-1})$. Hence, the expected total squared loss of the light rows is at least:

$$
(n - s - k)(1 - \mathcal{O}(n^{-1})) \geq (n - (1 + \alpha) \cdot k) - \mathcal{O}(n^{-1})
$$

Hence, the expected squared loss of a CountSketch whose sparsity is picked at random is at least

$$
\frac{\ell k}{4e} - \mathcal{O}(n^{-1}) + n - (1 + \alpha)k - \mathcal{O}(n^{-1}) \geq n + \frac{\ell k}{4e} - (1 + \alpha)k - \mathcal{O}(n^{-1})
$$

$\qquad \square$

**Corollary F.16.** *Let $s = \alpha(k-1)$ where $0.7 < \alpha < 1$ and let $\ell \geq \frac{(4e+1)n}{\alpha k}$. Let $S_g$ be the CountSketch whose sparsity pattern is learned over a training set drawn from $\mathcal{A}_{sp}$ via the greedy approach. Let $S_r$ be a CountSketch whose sparsity pattern is picked uniformly at random. Then, for an $n \times d$ matrix $A \sim \mathcal{A}_{sp}$ where $d = \Omega(n^6\ell^2)$, the expected loss of the best* rank-$k$ *approximation of $A$ returned by $S_r$ is worse than the approximation loss of the best* rank-$k$ *approximation of $A$ returned by $S_g$ by at least a constant factor.*

*Proof.*

$$\mathbf{E}\big[\min_{S_r \text{ rank-}k\ X \in \text{rowsp}(S_r A)} \|X - A\|_F^2\big] \geq n + \frac{\ell k}{4e} - (1 + \alpha)k - n^{-\mathcal{O}(1)} \qquad \rhd \text{ Lemma F.14}$$

$$\geq (1 + 1/\alpha)(n - s) \qquad \rhd \ell \geq \frac{(4e + 1)n}{\alpha k}$$

$$= (1 + 1/\alpha) \min_{\text{rank-}k\ X \in \text{rowsp}(S_g A)} \|X - A\|_F^2 \qquad \rhd \text{ Corollary F.13}$$

$\square$

### F.2 Zipfian on squared row norms.

Each matrix $A \in \mathbb{R}^{n \times d} \sim \mathcal{A}_{zipf}$ has rows which are uniformly random and orthogonal. Each $A$ has $2^{i+1}$ rows of squared norm $n^2/2^{2i}$ for $i \in [1, \ldots, \mathcal{O}(\log(n))]$. We also assume that each row has the same squared norm for all members of $\mathcal{A}_{zipf}$.

In this section, the $s$ rows with largest norm are called the *heavy* rows and the remaining are the *light* rows. For convenience, we number the heavy rows $1, \ldots, s$; however, the heavy rows can appear at any indices, as long as any row of a given index has the same norm for all members of $\mathcal{A}_{zipf}$. Also, we assume that $s \leq k/2$ and, for simplicity, $s = \sum_{i=1}^{h_s} 2^{i+1}$ for some $h_s \in \mathbb{Z}^+$. That means the minimum squared norm of a heavy row is $n^2/2^{2h_s}$ and the maximum squared norm of a light row is $n^2/2^{2h_s+2}$.

The analysis of the greedy algorithm ordered by non-increasing row norms on this family of matrices is similar to our analysis for the spiked covariance model. Here we analyze the case in which rows are orthogonal. By continuity, if the rows are close enough to being orthogonal, all decisions made by the greedy algorithm will be the same.

As a first step, by Lemma F.8, at the end of iteration $k$ the first $k$ rows are assigned to different bins. Then, via a similar inductive proof, we show that none of the light rows are mapped to a bin that contains one of the top $s$ heavy rows.

**Lemma F.17.** *At each iteration $k + r$, the greedy algorithm picks the position of the non-zero value in the $(k + r)$-th column of the CountSketch matrix $S$ so that the light row $A_{k+r}$ is mapped to a bin that does not contain any of top $s$ heavy rows.*

*Proof.* We prove the statement by induction. The base case $r = 0$ trivially holds as the first $k$ rows are assigned to distinct bins. Next we assume that in none of the first $k + r - 1$ iterations a light row is assigned to a bin that contains a heavy row. Now, we consider the following cases:

**1. If $A_{k+r}$ is assigned to a bin that only contains light rows.** Without loss of generality we can assume that $A_{k+r}$ is assigned to $b_k$. Since the vectors are orthogonal, we only need to bound the difference in the projection of $A_{k+r}$ and the light rows that are assigned to $b_k$ onto the direction of $w_k$ before and after adding $A_{k+r}$ to $b_k$. In this case, the total squared loss corresponding to rows in $b_k$ and $A_{k+r}$ before and after adding $A_{k+1}$ are respectively

$$\textit{before adding } A_{k+r} \textit{ to } b_k\text{: } \|A_{k+r}\|_2^2 + \sum_{A_j \in b_k} \|A_j\|_2^2 - \left(\frac{\sum_{A_j \in b_k} \|A_j\|_2^4}{\sum_{A_j \in b_k} \|A_j\|_2^2}\right)$$

$$\textit{after adding } A_{k+r} \textit{ to } b_k\text{: } \|A_{k+r}\|_2^2 + \sum_{A_j \in b_k} \|A_j\|_2^2 - \left(\frac{\|A_{k+r}\|_2^4 + \sum_{A_j \in b_k} \|A_j\|_2^4}{\|A_{k+r}\|_2^2 + \sum_{A_j \in b_k} \|A_j\|_2^2}\right)$$

Thus, the amount of increase in the squared loss is

$$
\begin{aligned}
\left(\frac{\sum_{A_j \in b_k} \|A_j\|_2^4}{\sum_{A_j \in b_k} \|A_j\|_2^2}\right) - \left(\frac{\|A_{k+r}\|_2^4 + \sum_{A_j \in b_k} \|A_j\|_2^4}{\|A_{k+r}\|_2^2 + \sum_{A_j \in b_k} \|A_j\|_2^2}\right) &= \frac{\|A_{k+r}\|_2^2 \cdot \sum_{A_j \in b_k} \|A_j\|_2^4 - \|A_{k+r}\|_2^4 \cdot \sum_{A_j \in b_k} \|A_j\|_2^2}{(\sum_{A_j \in b_k} \|A_j\|_2^2)(\|A_{k+r}\|_2^2 + \sum_{A_j \in b_k} \|A_j\|_2^2)} \\
&= \|A_{k+r}\|_2^2 \cdot \frac{\frac{\sum_{A_j \in b_k} \|A_j\|_2^4}{\sum_{A_j \in b_k} \|A_j\|_2^2} - \|A_{k+r}\|_2^2}{\sum_{A_j \in b_k} \|A_j\|_2^2 + \|A_{k+r}\|_2^2} \\
&\leq \|A_{k+r}\|_2^2 \cdot \frac{\sum_{A_j \in b_k} \|A_j\|_2^2 - \|A_{k+r}\|_2^2}{\sum_{A_j \in b_k} \|A_j\|_2^2 + \|A_{k+r}\|_2^2}
\end{aligned}
\tag{F.6}
$$

**2. If $A_{k+r}$ is assigned to a bin that contains a heavy row.** Without loss of generality and by the induction hypothesis, we assume that $A_{k+r}$ is assigned to a bin $b$ that only contains a heavy row $A_j$. Since the rows are orthogonal, we only need to bound the difference in the projection of $A_{k+r}$ and $A_j$ In this case, the total squared loss corresponding to $A_j$ and $A_{k+r}$ before and after adding $A_{k+1}$ to $b$ are respectively

*before adding $A_{k+r}$ to $b_k$:* $\|A_{k+r}\|_2^2$

*after adding $A_{k+r}$ to $b_k$:* $\|A_{k+r}\|_2^2 + \|A_j\|_2^2 - \left(\frac{\|A_{k+r}\|_2^4 + \|A_j\|_2^4}{\|A_{k+r}\|_2^2 + \|A_j\|_2^2}\right)$

Thus, the amount of increase in the squared loss is

$$
\|A_j\|_2^2 - \left(\frac{\|A_{k+r}\|_2^4 + \|A_j\|_2^4}{\|A_{k+r}\|_2^2 + \|A_j\|_2^2}\right) = \|A_{k+r}\|_2^2 \cdot \frac{\|A_j\|_2^2 - \|A_{k+r}\|_2^2}{\|A_j\|_2^2 + \|A_{k+r}\|_2^2}
\tag{F.7}
$$

Then (F.7) is larger than (F.6) if $\|A_j\|_2^2 \geq \sum_{A_i \in b_k} \|A_i\|_2^2$. Next, we show that at every inductive iteration, there exists a bin $b$ which only contains light rows and whose squared norm is smaller than the squared norm of any heavy row. For each value $m$, define $h_m$ so that $m = \sum_{i=1}^{h_m} 2^{i+1} = 2^{h_m+2} - 2$.

Recall that all heavy rows have squared norm at least $\frac{n^2}{2^{2h_s}}$. There must be a bin $b$ that only contains light rows and has squared norm at most

$$
\begin{aligned}
\|w\|_2^2 = \sum_{A_i \in b} \|A_i\|_2^2 &\leq \frac{n^2}{2^{2(h_s+1)}} + \frac{\sum_{i=h_k+1}^{h_n} \frac{2^{i+1}n^2}{2^{2i}}}{k-s} \\
&\leq \frac{n^2}{2^{2(h_s+1)}} + \frac{2n^2}{2^{h_k}(k-s)} \\
&\leq \frac{n^2}{2^{2(h_s+1)}} + \frac{n^2}{2^{2h_k}} &\quad \triangleright s \leq k/2 \text{ and } k > 2^{h_k+1} \\
&\leq \frac{n^2}{2^{2h_s+1}} &\quad \triangleright h_k \geq h_s + 1 \\
&< \|A_s\|_2^2
\end{aligned}
$$

Hence, the greedy algorithm will map $A_{k+r}$ to a bin that only contains light rows. $\qquad\square$

**Corollary F.18.** *The squared loss of the best* rank-$k$ *approximate solution in the rowspace of $S_g A$ for $A \in \mathbb{R}^{n \times d} \sim \mathcal{A}_{zipf}$ where $A \in \mathbb{R}^{n \times d}$ and $S_g$ is the CountSketch constructed by the greedy algorithm with non-increasing order, is* $< \frac{n^2}{2^{h_k-2}}$.

*Proof.* At the end of iteration $k$, the total squared loss is $\sum_{i=h_k+1}^{h_n} 2^{i+1} \cdot \frac{n^2}{2^{2i}}$. After that, in each iteration $k+r$, by (F.6), the squared loss increases by at most $\|A_{k+r}\|_2^2$. Hence, the total squared

loss in the solution returned by $S_g$ is at most

$$2\left(\sum_{i=h_k+1}^{h_n}\frac{2^{i+1}n^2}{2^{2i}}\right) = 4n^2 \cdot \sum_{i=h_k+1}^{h_n}\frac{1}{2^i} < \frac{4n^2}{2^{h_k}} = \frac{n^2}{2^{h_k-2}}$$

$\square$

Next, we bound the squared loss of the best $\text{rank-}k$-approximate solution constructed by the standard CountSketch with a randomly chosen sparsity pattern.

*Observation* F.19. Let us assume that the orthogonal rows $A_{r_1}, \cdots, A_{r_c}$ are mapped to the same bin and for each $i \leq c$, $\|A_{r_1}\|_2^2 \geq \|A_{r_i}\|_2^2$. Then, the total squared loss of $A_{r_1}, \cdots, A_{r_c}$ after projecting onto $A_{r_1} \pm \cdots \pm A_{r_c}$ is at least $\|A_{r_2}\|_2^2 + \cdots + \|A_{r_c}\|_2^2$.

*Proof.* Note that since $A_{r_1}, \cdots, A_{r_c}$ are orthogonal, for each $i \leq c$, the squared projection of $A_{r_i}$ onto $A_{r_1} \pm \cdots \pm A_{r_c}$ is $\|A_{r_i}\|_2^4 / \sum_{j=1}^{c}\|A_{r_j}\|_2^2$. Hence, the sum of squared projection coefficients of $A_{r_1}, \cdots, A_{r_c}$ onto $A_{r_1} \pm \cdots \pm A_{r_c}$ is

$$\frac{\sum_{j=1}^{c}\|A_{r_j}\|_2^4}{\sum_{j=1}^{c}\|A_{r_j}\|_2^2} \leq \|A_{r_1}\|_2^2$$

Hence, the total projection loss of $A_{r_1}, \cdots, A_{r_c}$ onto $A_{r_1} \pm \cdots \pm A_{r_c}$ is at least

$$\sum_{j=1}^{c}\|A_{r_j}\|_2^2 - \|A_{r_1}\|_2^2 = \|A_{r_2}\|_2^2 + \cdots + \|A_{r_c}\|_2^2.$$

$\square$

In particular, Observation F.19 implies that whenever two rows are mapped into the same bin, the squared norm of the row with smaller norm *fully* contributes to the total squared loss of the solution.

**Lemma F.20.** *For $k > 2^{10} - 2$, the expected squared loss of the best $\text{rank-}k$ approximate solution in the rowspace of $S_r A$ for $A_{n \times d} \sim \mathcal{A}_{zipf}$, where $S_r$ is the sparsity pattern of a CountSketch chosen uniformly at random, is at least $\frac{1.095 n^2}{2^{h_k-2}}$.*

*Proof.* In light of Observation F.19, we need to compute the expected number of collisions between rows with "large" norm. We can interpret the randomized construction of the CountSketch as a "balls and bins" experiment.

For each $0 \leq j \leq h_k$, let $\mathcal{A}_j$ denote the set of rows with squared norm $\frac{n^2}{2^{2(h_k-j)}}$ and let $\mathcal{A}_{>j} = \bigcup_{j < i \leq h_k} \mathcal{A}_i$. Note that for each $j$, $|\mathcal{A}_j| = 2^{h_k-j+1}$ and $|\mathcal{A}_{>j}| = \sum_{i=j+1}^{h_k} 2^{h_k-i+1} = \sum_{i=1}^{h_k-j} 2^i = 2(2^{h_k-j} - 1)$. Moreover, note that $k = 2(2^{h_k+1} - 1)$. Next, for a row $A_r$ in $\mathcal{A}_j$ ($0 \leq j < h_k$), we compute the probability that at least one row in $\mathcal{A}_{>j}$ collides with $A_r$.

$$\mathbf{Pr}[\text{at least one row in } \mathcal{A}_{>j} \text{ collides with } A_r] = \left(1 - \left(1 - \frac{1}{k}\right)^{|\mathcal{A}_{>j}|}\right)$$

$$\geq \left(1 - e^{-\frac{|\mathcal{A}_{>j}|}{k}}\right)$$

$$= \left(1 - e^{-\frac{2^{h_k-j}-1}{2^{h_k+1}-1}}\right)$$

$$\geq \left(1 - e^{-2^{-j-2}}\right) \qquad \triangleright \text{ since } \frac{2^{h_k-j}-1}{2^{h_k+1}-1} > 2^{-j-2}$$

Hence, by Observation F.19, the contribution of rows in $\mathcal{A}_j$ to the total squared loss is at least

$$(1 - e^{-2^{-j-2}}) \cdot |\mathcal{A}_j| \cdot \frac{n^2}{2^{2(h_k-j)}} = (1 - e^{-2^{-j-2}}) \cdot \frac{n^2}{2^{h_k-j-1}} = (1 - e^{-2^{-j-2}}) \cdot \frac{n^2}{2^{h_k-2}} \cdot 2^{j-1}$$

Thus, the contribution of rows with "large" squared norm, i.e., $\mathcal{A}_{>0}$, to the total squared loss is at least[4]

$$\frac{n^2}{2^{h_k-2}} \cdot \sum_{j=0}^{h_k} 2^{j-1} \cdot (1 - e^{-2^{-j-2}}) \geq 1.095 \cdot \frac{n^2}{2^{h_k-2}} \qquad \rhd \text{for } h_k > 8$$

$\square$

**Corollary F.21.** *Let $S_g$ be a CountSketch whose sparsity pattern is learned over a training set drawn from $\mathcal{A}_{sp}$ via the greedy approach. Let $S_r$ be a CountSketch whose sparsity pattern is picked uniformly at random. Then, for an $n \times d$ matrix $A \sim \mathcal{A}_{zipf}$, for a sufficiently large value of $k$, the expected loss of the best $\mathrm{rank}$-$k$ approximation of $A$ returned by $S_r$ is worse than the approximation loss of the best $\mathrm{rank}$-$k$ approximation of $A$ returned by $S_g$ by at least a constant factor.*

*Proof.* The proof follows from Lemma F.20 and Corollary F.18. $\square$

*Remark* F.22. We have provided evidence that the greedy algorithm that examines the rows of $A$ according to a non-increasing order of their norms (i.e., *greedy with non-increasing order*) results in a better $\mathrm{rank}$-$k$ solution compared to the CountSketch whose sparsity pattern is chosen at random. However, still other implementations of the greedy algorithm may result in a better solution compared to the greedy algorithm with non-increasing order. To give an example, in the following simple instance the greedy algorithm that checks the rows of $A$ in a random order (i.e., *greedy with random order*) achieves a $\mathrm{rank}$-$k$ solution whose cost is a constant factor better than the solution returned by the greedy with non-increasing order.

Let $A$ be a matrix with four orthogonal rows $u, u, v, w$ where $\|u\|_2 = 1$ and $\|v\|_2 = \|w\|_2 = 1 + \epsilon$ and suppose that the goal is to compute a $\mathrm{rank}$-$2$ approximation of $A$. Note that in the greedy algorithm with non-decreasing order, $v$ and $w$ will be assigned to different bins and by a simple calculation we can show that the copies of $u$ also will be assigned to different bins. Hence, the squared loss in the computed $\mathrm{rank}$-$2$ solution is $1 + \frac{(1+\epsilon)^2}{2+(1+\epsilon)^2}$. However, the optimal solution will assign $v$ and $w$ to one bin and the two copies of $u$ to the other bin which results in a squared loss of $(1 + \epsilon)^2$ which is a constant factor smaller than the solution returned by the greedy algorithm with non-increasing order for sufficiently small values of $\epsilon$.

On the other hand, in the greedy algorithm with a random order, with a constant probability of $(\frac{1}{3} + \frac{1}{8})$, the computed solution is the same as the optimal solution. Otherwise, the greedy algorithm with random order returns the same solution as the greedy algorithm with a non-increasing order. Hence, in expectation, the solution returned by the greedy with random order is better than the solution returned by the greedy algorithm with non-increasing order by a constant factor.

# G EXPERIMENT DETAILS

## G.1 LOW-RANK APPROXIMATION

In this section, we describe the experimental parameters in our experiments. We first introduce some parameters in Stage 2 of our approach proposed in Section 3.

- $bs$: batch size, the number of training samples used in one iteration.
- $lr$: learning rate of gradient descent.
- $iter$: the number of iterations of gradient descent.

**Table 7.1: Test errors for LRA (using Algorithm 2 with four sketches)**

For a given $m$, the dimensions of the four sketches were: $S \in \mathbb{R}^{m \times n}, R \in \mathbb{R}^{m \times d}, S_2 \in \mathbb{R}^{5m \times n}, R_2 \in \mathbb{R}^{5m \times d}$.

Parameters of the algorithm: $bs = 1, lr = 1.0, 10.0$ for hyper and video respectively, $num\_it = 1000$. For our algorithm 4, we use the average of all training matrix as the input to the algorithm.

---

[4]The numerical calculation is computed using WolframAlpha.

**Table 7.1: Test errors for LRA (using Algorithm 1 with one sketch)**

Parameters of the algorithm: $bs = 1, lr = 1.0, 10.0$ for hyper and video respectively, $num\_it = 1000$. For our algorithm 4, we use the sum of all training matrix as the input to the algorithm.

### G.2 Second-Order Optimization

As we state in Section 6, when we fix the positions of the non-zero entries (uniformly chosen in each column or sampled according to the heavy leverage score distribution), we aim to optimize the values by gradient descent, as mentioned in Section 3. Here the loss function is given in Section 6. In our implementation, we use PyTorch (Paszke et al. (2019)), which can compute the gradient automatically (here we can use torch.qr() and torch.svd() to define our loss function). For a more nuanced loss function, which may be beneficial, one can use the package released in Agrawal et al. (2019), where the authors studied the problem of computing the gradient of functions which involve the solution to certain convex optimization problem.

As mentioned in Section 2, each column of the sketch matrix $S$ has exactly one non-zero entry. Hence, the $i$-th coordinate of $p$ can be seen as the non-zero position of the $i$-th column of $S$. In the implementation, to sample $p$ randomly, we can sample a random integer in $\{1, \ldots, m\}$ for each coordinate of $p$. For the heavy rows mentioned in Section 6, we can allocate positions $1, \ldots, k$ to the $k$ heavy rows, and for the other rows, we randomly sample an integer in $\{k + 1, \ldots, m\}$. We note that once the vector $p$, which contains the information of the non-zero positions in each column of $S$, is chosen, it will not be changed during the optimization process in Section 3.

Next, we introduce the parameters for our experiments:

- $bs$: batch size, the number of training samples used in one iteration.
- $lr$: learning rate of gradient descent.
- $iter$: the number of iterations of gradient descent

In our experiments, we set $bs = 20, iter = 1000$ for all datasets. We set $lr = 0.1$ for the Electric dataset.

