# OpenReview forum: "Learning the Positions in CountSketch"
_ICLR.cc/2023/Conference — ICLR 2023 notable top 25%_

### Official Review · Reviewer_oTN6 · 2022-10-23

**Confidence:** 2
**Clarity, Quality, Novelty And Reproducibility:** Good clarity, quality and novelty. I …
**Correctness:** 4
**Technical Novelty And Significance:** 4
**Empirical Novelty And Significance:** 4
**Recommendation:** 8

**Strength And Weaknesses:**

Strength
1. Learning the position of non-zero entries makes sense and is a good idea.
2. The two algorithms are simple and thus practical, and come with theoretical guarantee.
3. The algorithms significantly reduce the approximation errors in the experiments.

Weakness
Nothing that I can think of.


**Summary Of The Paper:**

This paper proposes to learn the positions of the no-zero entries in count-sketches for better approximation accuracy. It improves previous approaches learn only the value of the no-zero entries. It first proposes a greedy solution, which has high complexity, then designs simpler algorithms for low rank approximation and Hessian approximation. Theoretical analysis shows that the algorithms reduce the size of the sketch without harming worst case performance.

**Summary Of The Review:**

This paper proposes to learn the positions of non-zero entries in count-sketches. The proposed methods are practical and come with strong theoretical guarantees. The empirical results demonstrate the effectiveness of the proposed algorithms.

---

> ### Author Response · Authors · 2022-11-14
> **Response to Reviewer oTN6**
>
> We thank the reviewer for the encouraging comments. We have reorganized some parts of our paper to make the presentation clearer (please refer to the revised version for details).

---

### Official Review · Reviewer_Akjx · 2022-10-24

**Confidence:** 2
**Correctness:** 3
**Technical Novelty And Significance:** 3
**Empirical Novelty And Significance:** Not applicable
**Recommendation:** 6

**Clarity, Quality, Novelty And Reproducibility:**

The work seems novel ( i have not read a lot of recent papers in this field. so i am not sure).

**Strength And Weaknesses:**

[Strengths] The paper is well written and relatively easy to understand (except for some clarification needed. see above) . Intuitions are provided with associated theory which is very much appreaciated.
[weakness] comparisions where only pure-sketching  methods are considered and the sampling methods such as leverage score sampling which is used by the algorithm itself is not used as a baseline.


**Summary Of The Paper:**

The paper focuses on sketching for linear algebraic problems like LRA. They move the state-of-the-art boundary theoretically and support it empirically. The main idea is to learn positions of non-zeros in the countsketch matrix . The approach seems reasonable and as shown in experiments is valuable.  Some comments /questions

1. not a learning heavy paper. Crux of contributing algorithms do not use learning and its target problems as well are linear. So not sure if it fits the call of papers. Something for Meta-Reviewers / Area chairs to decide.

2. The setting is not clear. where there is a set of training matrices {A1, .. An} and the test is supposed to be on an unseen data from the same distribution. This is not very clear. For example when you say A \in \mathcal{D}, does it mean the order of rows in A is also important? Or does it mean "each row of A" \in \mathcal{D}. In the latter case, it is difficult ot imagine how a fixed S will work for unseen A. For example S learned on A will not work on a permutation of A.  Maybe it will help clarifying this better.

3. I have not read much literature on learning of sketching matrices. so maybe its well known. I think it is important to point out the downside of these approaches. For example, the CS matrix is O(1) storage due to use of universal functions, however the storage of learned sketch matrix requires O(n) memory. It is important to point out this trade-off between CS vs learned sketch

4. The proposed approach is compared with sketching baselines namely - vanilla CS, Indyk 2019. However, as the authors also use existing techniques like ridge-leverage score sampling etc. It seems important to evaluate a comparison between sampling approaches (which in some sense are sketching as well). For example, a comparison with cohen 2017 - both theoretical and empirical would be good to have.

**Summary Of The Review:**

I believe the paper is a good contribution to the field. I would like resolutions to some of the issues with writing and experiments.

---

> ### Author Response · Authors · 2022-11-14
> **Response to Reviewer Ajkx**
>
> We thank the reviewer for their thoughtful comments.
>
> 1. The topic of our paper.
>
> The setting of our paper assumes that the underlying data matrix comes from some unknown distribution (e.g., video, data logs, customer activity, etc.), and we aim to use historical samples of the data to learn better sketching matrices than the classical CountSketch, which is oblivious to the data. All of our learned sketches are derived from the training matrix. Hence, we think our algorithms fit squarely into the learning-augmented algorithms paradigm. A number of learning-augmented algorithms have been accepted to ICLR recently, the following being some examples:
>
> - Triangle and Four Cycle Counting with Predictions in Graph Streams. ICLR'22 (https://openreview.net/forum?id=8in_5gN9I0)
> - Online Facility Location with Predictions. ICLR'22 (https://openreview.net/forum?id=DSQHjibtgKR)
> - Learning-Augmented k-means Clustering. ICLR'22(https://openreview.net/forum?id=X8cLTHexYyY)
> - Learning-Based Support Estimation in Sublinear Time. ICLR'21 (https://openreview.net/pdf?id=tilovEHA3YS)
> - Learning-Augmented Data Stream Algorithms. ICLR'20 (https://openreview.net/forum?id=HyxJ1xBYDH)
> - Learning-Based Frequency Estimation Algorithms. ICLR'19 (https://openreview.net/forum?id=r1lohoCqY7)
>
>
> 2. The setting of our paper.
>
> We assume that $\mathcal{D}$ is an unknown distribution over $n\times d$ matrices and $A$ is sampled from $\mathcal{D}$. Specifically, in the training set $\\mathsf{Tr} = \\{A_1, \ldots, A_N\\}$, each $A_i \in \mathbb{R}^{n \times d}$ is sampled from $\mathcal{D}$. It is therefore not to be expected that our learned sketch will work very well if one wildly permutes the rows of the input matrix. However, as we have shown in Appendix B.1, sometimes even if the rows are permuted, we can still recognize the pattern of the input matrix quickly and still gain a significant improvement over oblivious sketches such as CountSketch.
>
> 3. Storing the learned sketch.
>
> Thanks for bringing up this interesting point. It is true that $\Theta(n)$ space is required to store our learned CountSketch, as well as the learned sketches in previous work on sketching for optimization. However, notice that in the streaming model, the low-rank approximation problem already requires $\Omega(k \epsilon
> ^{-1}(n + d))$ memory (see, e.g., [CW09]), which is much larger than the space for storing the sketching matrix itself. We also note that the streaming model is not our main focus here; rather our focus is on accuracy and runtime.
>
> 4. Comparison to sampling-based approaches.
>
> We give a sensitivity analysis of our second approach in Appendix A.1, where some of the baselines we use are sampling-based approaches. Our result shows that naively using a sampling-based approach on the training data does not yield better results on the test data than a random CountSketch.

---

### Official Review · Reviewer_ZY3u · 2022-10-27

**Confidence:** 4
**Clarity, Quality, Novelty And Reproducibility:** The paper is well-written and all cla…
**Correctness:** 4
**Technical Novelty And Significance:** 4
**Empirical Novelty And Significance:** 3
**Recommendation:** 8

**Strength And Weaknesses:**

Strengths:

+ The paper tackles a natural drawback of existing approaches for learning-based sketching algorithms. It provides good intuitions regarding the central premise of learning positions of non-zero entries of the sketching matrix as opposed to just their values (as prior work)

+ The paper is well-written and all claims are well justified. The paper is well-written and includes detailed algorithms for clarity. I feel the presentation of the paper got hurt significantly due to page restrictions, and I encourage the authors to publish a full version where the flow of text is not hurt due to page limits.


**Summary Of The Paper:**

The paper considers sketching based algorithms for low rank approximation and second-order methods for regression where the sketching matrix is learned. If the underlying data matrix is drawn from some unknown distribution, one can use historical samples of the data to learn better sketch matrices rather than the classical, data independent Count Sketch. While prior learning-based sketches only learn the values of non-zero entries of the sketching matrix (and keep the positions random), the paper proposes a framework to also learn the positions of the non-zero entries of the sketching matrix. The proposed algorithm is fairly natural and outperforms prior learned sketching based approaches.

**Summary Of The Review:**

Overall I feel the paper makes an important contribution. The key ideas introduced are fairly natural and many of the proofs follow from standard techniques. Empirical results illustrate that learning positions of non-zeros along with their values significantly improves the approximation of classical CountSketch (at a similar sketch size).

---

> ### Author Response · Authors · 2022-11-14
> **Response to Reviewer ZY3u**
>
> We thank the reviewer for the encouraging comments. We have reorganized some parts of our paper to make the presentation clearer (please refer to the revised version for details). Also, we will make a full version of our paper available.

---

### Official Review · Reviewer_qwvC · 2022-10-28

**Confidence:** 4
**Correctness:** 4
**Technical Novelty And Significance:** 3
**Empirical Novelty And Significance:** 3
**Recommendation:** 8

**Clarity, Quality, Novelty And Reproducibility:**

Clarity: The paper seems to be put together hurriedly. Presentation and flow can be improved quite a lot. See above for specific comments.

Quality: Idea is quite interesting and warrants a study. The paper is technically sound.

Originality: The idea is original, although the proposed algorithms rely quite heavily on existing frameworks (ridge leverage score sampling and sketch monotonicity).



**Strength And Weaknesses:**

Strengths:

1. The idea of learning both the positions and the values is interesting and warrants a study. This paper does a good job of exploring this (previously unexplored) idea.

2. The paper shows strong improvement over previous algorithms in terms of numerical performance.

Weaknesses:

1. The presentation of the paper can be improved a lot. Some examples of poor presentation are:
    - Algorithms 1, 2 and 3 are placed in unnatural locations, with presumably liberal usage of \vspace and \hspace. The paper is visually
        very dense and doesn't need to be.
    - On page 3: "Hence, in this work we consider the following algorithm that compresses both sides of A...": which algorithm?
    - Authors should consider completely moving the few-shot learning setting to the appendix and write more clearly about the two main
        problems considered.
    - Placement of the results tables are also not great. For example, Table 7.4 appear at a random location in between text.

2. I have the following questions regarding the experiments:
    - Shouldn't metrics always be relative errors? For example, in the low-rank approximation setting, instead of reporting ||A-A_k|| - ||A-X*||,
        it should be ( ||A-A_k|| - ||A-X*||)/ ||A||. This would indicate how large the error is w.r.t the Frobenius norm of A itself and gives a better
        idea of performance.
    - Why are the errors in the few-shot learning case smaller than when the full training set is used? Intuitively, it should be the opposite.
    - The authors mention that during training, only the average of the training matrices are used. However, this is not equivalent to
        minimizing the expected error over the training set. It is also not clear if the baseline methods (especially IVY19) follow this procedure.
        Upon reading the IVY19 paper, it appears that they do minimize the error over all the training matrices. Maybe I am missing something,
        but there is chance that the baseline methods have worse performance because of this? It would be better to present results (at least
        on the baseline methods) where the training fully takes advantage of the training set and does not use just the average.
    - Authors mention that they use torch.qr for backpropagating through QR factorization. Do the authors take any measures to ensure
       that the first few columns are linearly independent (as required by torch.qr: https://pytorch.org/docs/stable/generated/torch.qr.html)?


**Summary Of The Paper:**

This paper is about learning linear maps for dimensionality reduction that are based on count sketch. Existing works either use randomized sketches or learn the values in a count sketch matrix (with the positions drawn randomly). This paper develops on such methods by proposing to learn both the positions and the values of the count sketch matrix. The paper studies this idea in the context of two problems: low-rank approximation and iterative Hessian sketching. The contributions of the paper are:
1. It shows that learning positions and values instead of just values leads to better performance. This is shown using a greedy search algorithm
2. In order to overcome the high run time of the greedy search algorithm, the paper proposed algorithms to construct the count sketch matrix using ridge leverage score based sampling.
3. The paper shows using experiments that the proposed ideas lead to better numerical performance.

**Summary Of The Review:**

The paper studies an interesting problem and presents good ideas. However, the presentation needs to be improved and the experiments should be explained more clearly. I have elaborated further in my comments above.

---

> ### Author Response · Authors · 2022-11-14
> **Response to Reviewer qwvC**
>
> We thank the reviewer for their thoughtful comments.
>
> 1. Regarding the presentation issues:
>
> Thank you for the suggestion about the paper organization. We have moved the few-shot setting experiments to the appendix and put the algorithms and tables in their corresponding places. On page 3, we have moved the description of the two-sided sketching algorithm (Algorithm 2) to the main body of the paper. Please refer to the revised version for details.
>
> 2. Detailed questions about experiments.
>
> (1) We agree that the relative error $(\\|A-X\\|_F - \\|A - A_k\\|_F) / \\|A - A_k\\|_F$ is more natural. Here we choose an additive error metric to stay consistent with the setting considered in [IVY19]. We normalized the data used in the experiments. For reference, the average errors of the best rank-$k$ approximation $\\|A - A_k\\|_F$ when $k = 30$ for the Logo, Friends, and Hyper data are 3.52, 6.195, and 13.56, respectively.
>
> (2) The choice of parameters ($m, k$) in our experiments is different for the few-shot setting. Here $m = 40, k = 10$, which are the parameters chosen in the work of the previous few-shot paper (IWW21). Here $m$ is four times larger than $k$, so the error is smaller.
>
> (3) For the low rank approximation (LRA)  experiments, only our Algorithm 4 uses the average of the training matrices. The greedy algorithm and [IVY19] use the whole training set. We have emphasized this in our revised version.
>
> (4) In our sketching for second-order optimization, we assume that $n$ is much larger than $d$ and that the matrix $A$ has full column rank. Note that if $A$ does not have full column rank, we can remove linearly dependent columns (which can be found by either a standard QR decomposition algorithm, or faster input-sparsity time sketching-based methods), and obtain a matrix $A'$, which now has full column rank and the same column space as $A$. Then we have that if $S$ is a $(1 \pm \epsilon)$-subspace embedding for $A'$, then $S$ is also a $(1 \pm \epsilon)$-subspace embedding for $A$.

---

> > ### Comment · Reviewer_qwvC · 2022-11-16
> > **Updated score**
> >
> > I thank the authors for the clarifications provided. I am changing my score to 8.

---

### Decision · Program_Chairs · 2023-01-20

**Decision:**

Accept: notable-top-25%

**Justification For Why Not Higher Score:**

- There are a few points that are/were unclear in the paper (cf. computational space/savings, link to the literature)
- It's a paper that is not 100% centered on Machine Learning, but rather on data processing and they way to compact them -- which is still an ML problem, though.

**Justification For Why Not Lower Score:**

The content is of high quality.

**Metareview: Summary, Strengths And Weaknesses:**

This paper introduces novel sketching based algorithms for low rank approximation and second-order methods for regression where the sketching matrix is learned. The authors develo such methods by proposing to learn both the positions and the values of the count sketch matrix, that is key to compute sketches.

This is a good paper, appraised by all the reviewers for:
- the novelty provided: on crafting a sketching algorithm that learns the positions and values of the sketching matrix at once
- the method(s) proposed are theoretically studied and proven to be advantageous if compared to the existing literature
- empirical evaluation of the method(s) are provided, that show their effectiveness

The weaknesses of the paper have been answered in the discussion
- presentation
- some computational / space complexity had to be clarified.

It is important for the authors to implement the changes that they propose to answer the various comments.


**Note From Pc:**

if the above contains the word "oral" or "spotlight" please see: "oral" presentation means -> notable-top-5% and "spotlight" means -> notable-top-25%. As stated in our emails, we are disassociating presentation type from AC recommendations